# Corticohippocampal circuit dysfunction in a mouse model of Dravet syndrome

Joanna Mattis[1], Ala Somarowthu[2], Kevin M Goff[3], Evan Jiang[2], Jina Yom[4], Nathaniel Sotuyo[3], Laura M Mcgarry[2], Huijie Feng[2], Keisuke Kaneko[2], Ethan M Goldberg[1,2,5]*

[1]Department of Neurology, The Perelman School of Medicine at The University of Pennsylvania, Philadelphia, United States; [2]Division of Neurology, Department of Pediatrics, The Children's Hospital of Philadelphia, Philadelphia, United States; [3]Neuroscience Graduate Group, The University of Pennsylvania Perelman School of Medicine, Philadelphia, United States; [4]College of Arts and Sciences, The University of Pennsylvania, Philadelphia, United States; [5]Department of Neuroscience, The Perelman School of Medicine at The University of Pennsylvania, Philadelphia, United States

**Abstract** Dravet syndrome (DS) is a neurodevelopmental disorder due to pathogenic variants in *SCN1A* encoding the Nav1.1 sodium channel subunit, characterized by treatment-resistant epilepsy, temperature-sensitive seizures, developmental delay/intellectual disability with features of autism spectrum disorder, and increased risk of sudden death. Convergent data suggest hippocampal dentate gyrus (DG) pathology in DS ($Scn1a^{+/-}$) mice. We performed two-photon calcium imaging in brain slice to uncover a profound dysfunction of filtering of perforant path input by DG in young adult $Scn1a^{+/-}$ mice. This was not due to dysfunction of DG parvalbumin inhibitory interneurons (PV-INs), which were only mildly impaired at this timepoint; however, we identified enhanced excitatory input to granule cells, suggesting that circuit dysfunction is due to excessive excitation rather than impaired inhibition. We confirmed that both optogenetic stimulation of entorhinal cortex and selective chemogenetic inhibition of DG PV-INs lowered seizure threshold in vivo in young adult $Scn1a^{+/-}$ mice. Optogenetic activation of PV-INs, on the other hand, normalized evoked responses in granule cells in vitro. These results establish the corticohippocampal circuit as a key locus of pathology in $Scn1a^{+/-}$ mice and suggest that PV-INs retain powerful inhibitory function and may be harnessed as a potential therapeutic approach toward seizure modulation.

*For correspondence: goldberge@chop.edu

**Competing interest:** The authors declare that no competing interests exist.

## Editor's evaluation

Recent work has shown that one of the major dogmas in epilepsy – that interneuron deficits underlie Dravet syndrome and maybe other epileptic encephalopathies – is overly simplistic. This manuscript makes a significant step forward with novel findings using ex vivo and in vivo experiments providing strong evidence that changes in excitatory connections in the corticohippocampal circuit contribute to mechanisms that drive epilepsy in Dravet syndrome.

## Introduction

Pathogenic variants in *SCN1A*, which encodes the voltage-gated sodium channel α subunit Nav1.1, cause a spectrum of epilepsies including Dravet syndrome (DS) (*Claes et al., 2001*), the most common developmental and epileptic encephalopathy. DS is characterized by treatment-resistant epilepsy, developmental delay/intellectual disability, features of or formal diagnosis of autism spectrum disorder,

and motor dysfunction (hypotonia, ataxia, gait impairment) (*Villas et al., 2017*). DS-associated *SCN1A* variants are due to loss of function leading to haploinsufficiency of Nav1.1. The heterozygous *Scn1a* mutant (*Scn1a$^{+/-}$*) mouse is a well-established preclinical model of DS that recapitulates key phenotypic features of the human condition (*Mistry et al., 2014*). When expressed on a 50:50 129S6:C57BL/6J genetic background, these mice exhibit spontaneous seizures beginning at approximately post-natal day (P) 18, and high rates of sudden unexpected death in epilepsy (SUDEP) (*Mistry et al., 2014*). These mice also exhibit temperature-sensitive seizures, akin to seizures triggered in the setting of fever or hyperthermia in human patients with DS, which represents a key experimental advantage of this mouse model as it readily facilitates study of inducible but naturalistic seizures in vivo (*Tran et al., 2020*).

The prevailing theory as to how reduction in sodium current leads to epilepsy in DS is the so-called 'interneuron hypothesis', which posits that haploinsufficiency of Nav1.1 results in selective deficits in inhibitory interneuron excitability based on the relative reliance of this cell class on Nav1.1 for action potential generation. Nav1.1 is prominently expressed in parvalbumin-expressing GABAergic inhibitory interneurons (PV-INs) in neocortex (*Ogiwara et al., 2007*), as well as by somatostatin (SST-INs) and vasoactive intestinal peptide-expressing interneurons (VIP-INs). Electrophysiological recordings from acutely dissociated hippocampal neurons from *Scn1a$^{+/-}$* mice found decreased sodium current density in bipolar-shaped presumptive GABAergic interneurons, but not in pyramidal cells (*Bechi et al., 2012*; *Yu et al., 2006*); impaired excitability of inhibitory interneurons – PV-INs as well as SST and VIP-INs – has been demonstrated in acute brain slices from *Scn1a$^{+/-}$* mice (*Favero et al., 2018*; *Goff and Goldberg, 2019*; *Ogiwara et al., 2007*; *Tai et al., 2014*). Consistent with this, human DS patient-derived induced pluripotent stem cells (iPSCs) differentiated to form developing interneurons with biophysical deficits, whereas derived excitatory neurons were normal (*Sun et al., 2016*). This 'interneuron hypothesis' is further supported by the fact that the DS phenotype is recapitulated by selective loss of Nav1.1 exclusively in GABAergic interneurons (*Cheah et al., 2012*; *Dutton et al., 2013*; *Ogiwara et al., 2013*; *Rubinstein et al., 2015*).

However, multiple lines of evidence suggest that the pathophysiology of epilepsy in *Scn1a$^{+/-}$* mice is more complex than impairment of inhibition. First, the net circuit effect of VIP-INs is disinhibition, so the abnormal excitability of this population (*Goff and Goldberg, 2019*) should in fact increase inhibition in neocortical circuits. Second, contradictory data from the *Scn1a$^{+/-}$* mouse models (*Mistry et al., 2014*) and from human-derived iPSCs (*Jiao et al., 2013*; *Liu et al., 2013*) suggest that subsets of excitatory neurons in DS may in fact be hyperexcitable. Third, recent work found that impaired action potential generation present at early developmental time points in neocortical PV-INs normalized by P35 (*Favero et al., 2018*), whereas DS mice continue to exhibit epilepsy, cognitive impairment, and SUDEP. Finally, several in vivo studies have failed to find a clear decrease in baseline interneuron activity in *Scn1a$^{+/-}$* mice (*De Stasi et al., 2016*; *Tran et al., 2020*).

Hence, mechanisms of seizure generation and maintenance of chronic epilepsy in DS remain unclear despite this now growing literature characterizing deficits in *Scn1a$^{+/-}$* mice at a single cell level (*Cheah et al., 2012*; *Dutton et al., 2013*; *Favero et al., 2018*; *Goff and Goldberg, 2019*; *Ogiwara et al., 2007*; *Tai et al., 2014*). This may be in part due to the lack of data linking cellular deficits to circuit-level abnormalities. Convergent data suggest that the dentate gyrus (DG) may be a key locus of pathology and seizure generation in *Scn1a$^{+/-}$* mice. Although *Scn1a$^{+/-}$* mice exhibit multifocal epilepsy, temperature-induced seizures have been shown to prominently emanate from the temporal lobe (*Liautard et al., 2013*), and focal hippocampal Nav1.1 reduction is sufficient to confer temperature-sensitive seizure susceptibility in conditional *Scn1a$^{+/-}$* mice (*Stein et al., 2019*). Although the DG receives a strong excitatory input from entorhinal cortex (*Amaral et al., 2007*), inhibition within the DG typically regulates population activity such that granule cells (GCs) are only sparsely activated under physiologic conditions both in reduced preparations in vitro and in experimental animals in vivo (*Chawla et al., 2005*; *Diamantaki et al., 2016*; *Dieni et al., 2013*; *Ewell and Jones, 2010*; *Lee et al., 2016*; *Liu et al., 2014*; *Neunuebel and Knierim, 2012*; *Senzai and Buzsáki, 2017*; *Yu et al., 2013*). However, seizure-evoked immediate early gene activation is most apparent in the DG granule cell layer in *Scn1a$^{+/-}$* mice (*Dutton et al., 2017*), suggesting an underlying circuit-level hyperexcitability.

In this study, we demonstrate a profound hyperexcitability of the corticohippocampal circuit in young adult *Scn1a$^{+/-}$* mice that is not present at epilepsy onset using two-photon calcium imaging and cellular and synaptic physiology in an acute slice preparation. We find this circuit dysfunction is likely

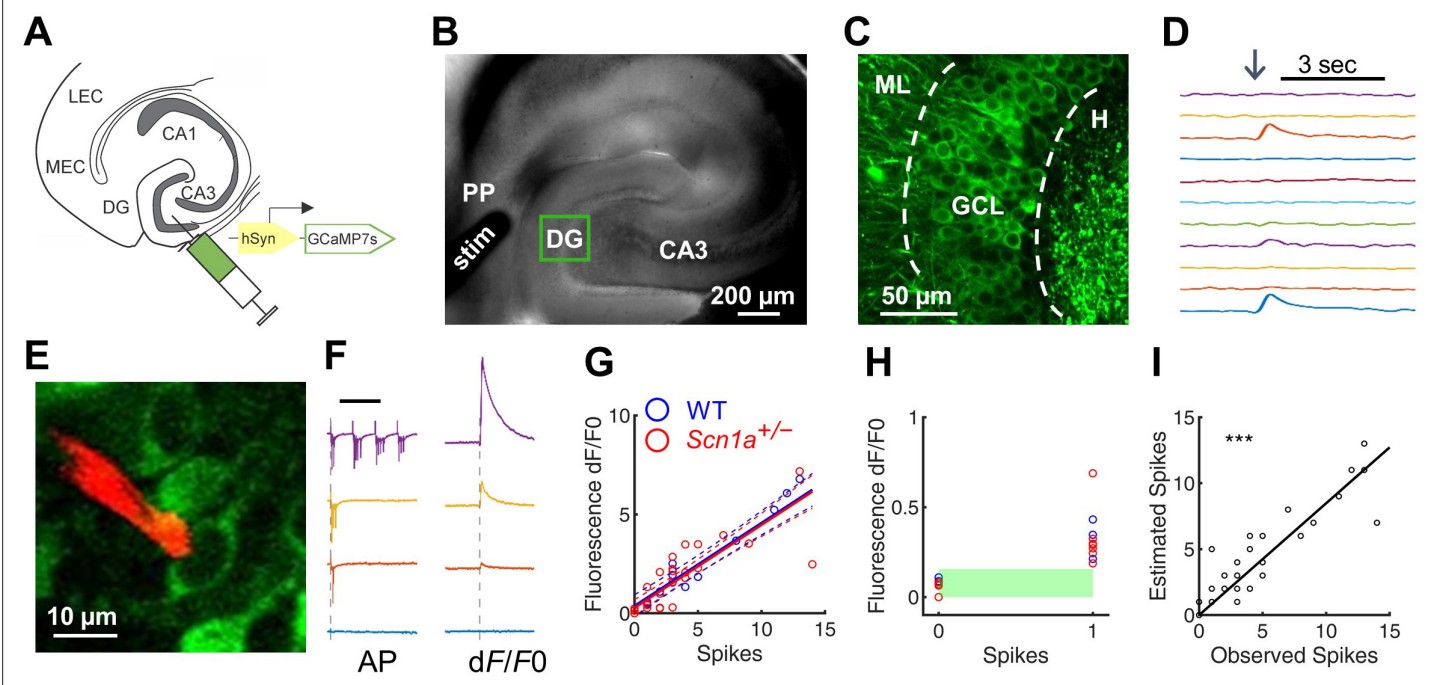

**Figure 1.** Two-photon calcium imaging of perforant path-evoked dentate gyrus activation in acute brain slice. (**A**) Mice were injected with AAV9-hSyn-jGCaMP7s-WPRE into DG. (**B**) Acute slices are cut at a 15° angle off-axial to maximize connectivity. Shown is the location of a stimulation electrode (*stim*) in the perforant path (PP), and the imaging field in DG (*green box*). (**C**) GCaMP labeling of GCs in the granule cell layer (GCL; *dashed lines*) between the hilus (H) and molecular layer (ML). (**D**) Example GC response to PP stimulation (*arrow*), displayed as change in fluorescence over baseline (d*F*/*F*0). (**E**) Representative image showing a quantum dot (***Andrásfalvy et al., 2014***)-labeled pipette tip (*red*) used for cell-attached recording of GCaMP-expressing GCs (*green*). (**F**) Representative data from a single GC firing 0, 1, 3, or 13 action potentials (*left*) in response to PP stimulation, while calcium transients (*right*) were simultaneously recorded. Scale bar represents 75 ms (*left*) and 5 s (*right*). Vertical dashed line indicates stimulus onset. (**G**) Action potentials versus calcium transient magnitude for GCs from WT (*blue*) and *Scn1a*$^{+/-}$ (*red*) mice. Data were fit using a linear model, with no significant difference between genotypes (solid / dashed line = best fit / 95% confidence interval). $n$ = 7 (cells), 5 (mice). (**H**) Results for 0 and 1 action potentials. Threshold for action potential detection (*green bar*) was defined as $p$ = 0.001 from a normal distribution fit to the d*F*/*F*0 values for 0 action potentials. (**I**) Observed action potentials versus action potentials derived from deconvolution ($R^2$ = 0.83; $p < 0.0001$).

The online version of this article includes the following source data for figure 1:

**Source data 1.** Quantification of spikes.

driven by excessive excitation, as opposed to impaired inhibition. We then extend these findings in vivo using optogenetics in awake, behaving mice during temperature-sensitive seizures to demonstrate a role for this corticohippocampal dysregulation in ictogenesis. Finally, we selectively inhibit DG PV-INs (via chemogenetics) to exacerbate seizures in vivo, and recruit PV-INs (via optogenetics) to decrease evoked DG granule cell response to perforant path input in brain slice in vitro. These findings highlight the corticohippocampal circuit as a critical locus of pathology in DS and suggest that PV-INs retain powerful regulation of excitability within the circuit.

## Results

### Response of dentate gyrus to entorhinal cortical input is selectively impaired in young adult *Scn1a+/-* mice

Emerging data suggest an important role for the hippocampus and, more specifically, the dentate gyrus (DG), in DS pathology (***Dutton et al., 2013***; ***Liautard et al., 2013***; ***Stein et al., 2019***). We therefore hypothesized that the circuit function of DG – filtering input from the entorhinal cortex to limit propagation of activity through the limbic system – might be impaired in *Scn1a*$^{+/-}$ mice. We used two photon calcium imaging (2 P imaging) to achieve large-scale quantification of granule cell (GC) activation in response to stimulation of the perforant path (PP), the excitatory projection from entorhinal cortex that constitutes the major input to GCs. We first injected an adeno-associated virus

(AAV) encoding the calcium indicator GCaMP7s (*Dana et al., 2019*) under control of a pan-neuronal promoter hSyn1 into DG of *Scn1a*[+/-] mice and age-matched wild-type littermate controls (*Figure 1A*). We prepared acute hippocampal-entorhinal cortex (HEC) slices (*Xiong et al., 2017*), which preserve the PP projection from entorhinal cortex to hippocampus within the slice (*Figure 1B*). We then performed 2 P imaging of evoked calcium transients in cells within the granule cell layer (GCL) in response to electrical stimulation of the PP (*Figure 1C–D*).

To characterize GCaMP7s as a reporter of action potentials in GCs within this experimental paradigm, we performed simultaneous cell-attached recording and calcium imaging of GCaMP-expressing cells within the GCL in *Scn1a*[+/-] and wild-type mice. We used a quantum dot-labeled pipette tip (*Andrásfalvy et al., 2014*) for 2P-guided targeted electrophysiological recording during simultaneous calcium imaging (*Figure 1E*). This allowed us to correlate calcium transients within individual cells against a gold-standard quantitative measure of action potentials (APs). Action potentials were evoked via PP stimulation. However, given the normally sparse activation of DG in response to PP stimulation in wild-type mice, we bath-applied picrotoxin (PTX), a GABA$_A$ receptor antagonist that is known to induce widespread PP-driven recruitment of GCs (*Dengler et al., 2017*), such that nearly all GCs were activated by PP stimulation. We found that the number of APs and the magnitude of the calcium signal were highly correlated (*Figure 1F–G*; $R^2 = 0.769$). This linear relationship between AP and calcium signal was the same in wild-type and *Scn1a*[+/-] mice for single APs and across a measured range (0–14 APs; *Figure 1G*). We next evaluated the sensitivity and specificity of GCaMP7s in detecting single APs in the absence of PTX as GABA$_A$-mediated responses in dentate GCs can be depolarizing even in the adult DG (*Chiang et al., 2012*), and could therefore cause subthreshold calcium transients which may mimic spikes (*Stocca et al., 2008*). We fit a normal distribution to the d$F$/$F0$ values associated with 0 APs (with an AP defined as an increase in calcium signal (d$F$/$F0$) of at least three times the standard deviation above the mean noise), set a global threshold defined by this data (p = 0.001), and verified that subthreshold responses (n = 9) could be resolved from single APs (n = 10; *Figure 1H*). Finally, we deconvolved the d$F$/$F0$ signals (*Pnevmatikakis et al., 2016*) to extract the number of action potentials from the calcium transients and verified the deconvolution algorithm on the 'ground truth' AP dataset ($R^2 = 0.83$; p < 0.0001; *Figure 1I*). Thus, GCaMP7s reports APs in GCs similarly in both genotypes, with single-AP resolution, validating 2 P calcium imaging in brain slice as a method for comparing large-scale DG excitability between wild-type and *Scn1a*[+/-].

In order to quantify the extent of DG activation in response to entorhinal cortex input, we stimulated the PP while performing 2 P imaging of the evoked responses across hundreds of GCs simultaneously (*Figure 2*). We tested early postnatal (P14-21; at/around epilepsy onset) and young adult (P50-100; chronic phase) mice. We delivered either a single pulse or a train of 4 pulses at 20 Hz across a range of stimulation intensities, similar to prior studies of this circuit in acute brain slices (*Dengler et al., 2017*; *Ewell and Jones, 2010*; *Yu et al., 2013*). We then identified activated GCs within the imaging field in response to each stimulation condition, defining activation as an increase in calcium signal (d$F$/$F0$) at least three times the standard deviation above the mean noise (*Figure 2*, columns 1–4) and/or if one or more action potentials were extracted using deconvolution (*Figure 2*, columns 5–6). To determine if the responses across stimulation intensities differed by genotype, we employed a statistical approach based on mixed-effects modeling, to account for potential variation between individual imaging fields in a slice, slices from a given mouse, and/or between mice. (See *Figure 2—figure supplement 1* for data including all individual imaging fields, which are omitted for clarity from *Figure 2*.)

We first calculated the proportion of activated GCs relative to all identifiable GCaMP7s-expressing cells, fit the data from each genotype (as a binomial distribution fitted with a probit link function), and compared the fit of the curves between genotypes ('All cells' in *Figure 2A*). We validated these data in two different ways to account for the possibility that some GCs might be unresponsive due to deafferentation during the slice preparation, as opposed to this being due to physiologic circuit inhibition. First, we excluded from analysis all GCs that failed to respond to maximal PP stimulation in the presence of PTX ('Responds to PTX' in *Figure 2B*). Note that most ( > 85%) GCs did respond under these conditions (*Figure 2—figure supplement 2*), demonstrating a high level of connectivity within the HEC slice; hence, results were similar whether considering all cells or only PTX-responsive cells. Second, we excluded from analysis those GCs that failed to respond to the highest amplitude PP stimulation delivered in the absence of PTX ('Responds to max stim' in *Figure 2C*). Finally, in *Figure 2D*,

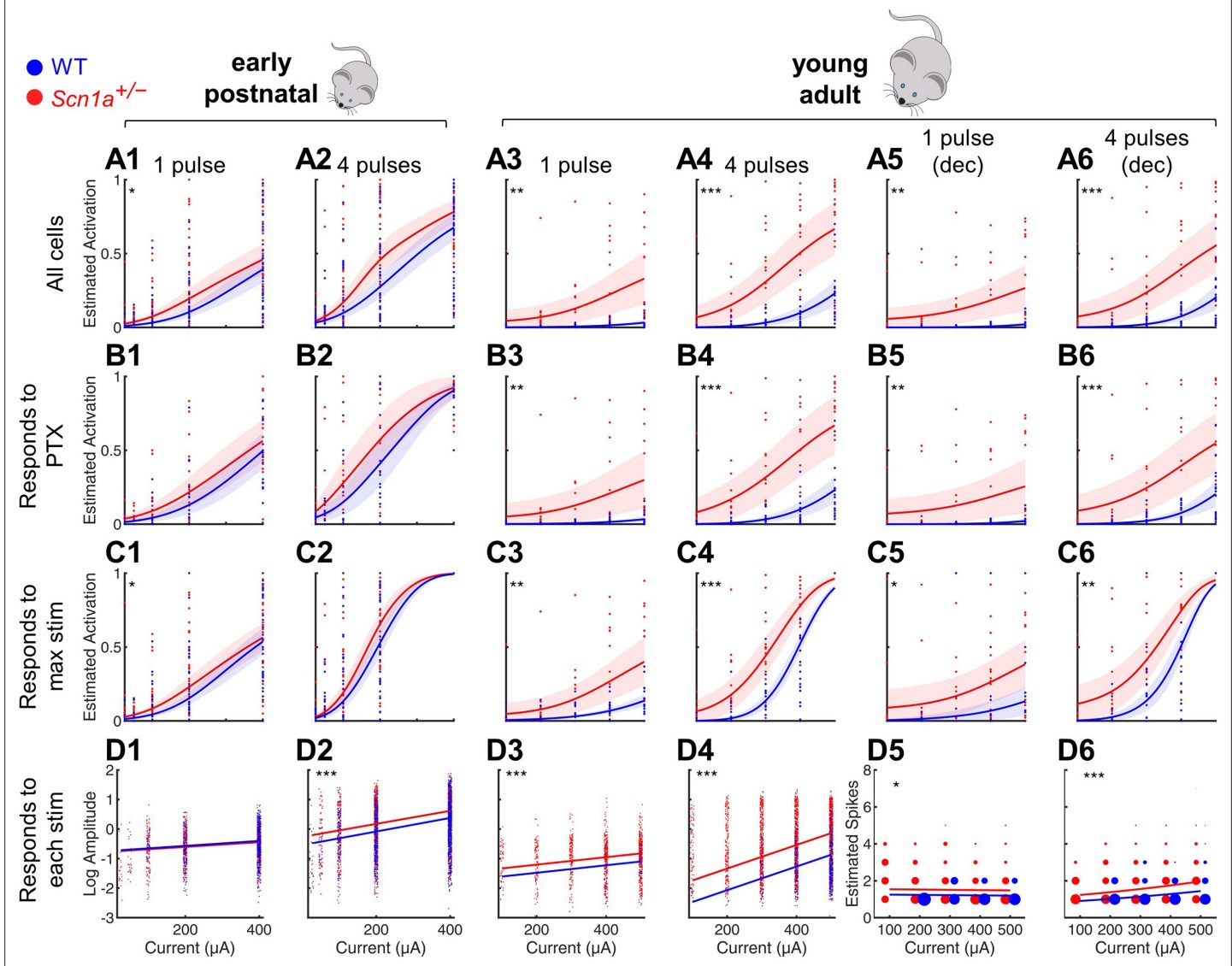

**Figure 2.** Selective impairment of dentate gyrus function in young adult *Scn1a*⁺/⁻ mice. Proportion of activated GCs (**A–C**) and magnitude of activation (**D**) in response to PP stimulation (early postnatal P14-21, columns 1 and 2; young adult P48-91, columns 3 and 4), with deconvolved data for the young adult time point (columns 5 and 6). Wild-type in *blue; Scn1a*⁺/⁻ in *red*. Data were analyzed using a mixed model to account for potential variability between animal, slice, field, and cell. Proportion of responsive GCs was calculated (**A**) relative to all GCaMP-expressing cells, or restricting analysis (**B**) to only GCs that respond to PP stimulation in the presence of 100 μM picrotoxin or (**C**) to only GCs that respond to the maximal stimulation delivered. (**D**) Magnitude of activated GC responses to PP stimulation, expressed as natural log d*F/F*0 (columns 1–4) or estimated spikes based on deconvolution (columns 5–6; size of data points reflects number of cells at that value). *Dots*: raw data from all imaging fields or cells; *dark lines*: average of fits; *shaded*: 95% confidence intervals of average fits. Stars indicate significant differences in curve fits: *, p < 0.05; **, p < 0.01; ***, p < 0.001. For early postnatal mice, *n* for the experiments in rows A, C, and D was, for *Scn1a*⁺/⁻ and WT, respectively: 991 and 1,210 (cells), 51 and 58 (fields), 31 and 29 (slices), 11 and 10 (mice); for row B: 372 and 445 (cells), 16 and 17 (both fields and slices), 5 and 5 (mice). For the young adult mice, *n* for the experiments in rows A, C, and D: 1236 and 1167 (cells), 17 and 17 (both fields and slices), 8 and 6 (mice); for row B: 1109 and 1073 (cells), 16 and 14 (both fields and slices), 6 and 7 (mice).

The online version of this article includes the following source data and figure supplement(s) for figure 2:

**Source data 1.** Quantified imaging data, reported by cell.

**Source data 2.** Quantified imaging data, reported by field.

**Source data 3.** Quantified imaging data, summarizing field response to PTX.

**Figure supplement 1.** Selective impairment of dentate gyrus function in young adult *Scn1a*⁺/⁻ mice.

**Figure supplement 2.** A high percentage of GCs are activated by PP stimulation in the presence of PTX.

*Figure 2 continued on next page*

*Figure 2 continued*

**Figure supplement 3.** Estimation of total evoked spikes in young adult GCs.

**Figure supplement 4.** *Scn1a*$^{+/-}$ DG GCs have normal firing properties at both early postnatal and young adult timepoints.

we quantified the magnitude of the evoked calcium signal (d$F$/$F$0; columns 1–4) or estimated the action potential numbers calculated via deconvolution (columns 5–6) and used a mixed-effects model to compare across genotypes, using a normal distribution fitted with a logarithmic link function. Note that we included in this analysis only GCs activated under each specific condition ('Responds to each stim'); that is we quantified how many spikes occurred per active cell. We additionally analyzed the data including all GCs activated under highest amplitude PP stimulation ('Responds to max stim'; *Figure 2—figure supplement 3*) to quantify the total activation of the population: that is how many spikes occurred, averaged across cells.

We saw enhanced activation at the early postnatal timepoint for both genotypes (*Figure 2*, columns 1–2), with only a subtle difference between genotypes. Consistent with the known increase in sparsity of GC activation that occurs with development (*Yu et al., 2013*), the wild-type activity pattern was markedly decreased at the young adult relative to the early postnatal timepoint (*Figure 2*, columns 3–6). However, the *Scn1a*$^{+/-}$ activation pattern remained markedly enhanced at the young adult relative to the early postnatal timepoint, resulting in a robust, statistically significant difference between genotypes at this later timepoint. For instance, proportional *Scn1a*$^{+/-}$ GC activation at the young adult timepoint was over threefold larger than that of wild-type (0.51 ± 0.08 vs 0.16 ± 0.00; p < 0.001) in response to 4 × 400 μA pulses (subset of data in *Figure 2A* 4), and d$F$/$F$0 was increased by ~130% (p < 0.001; *Figure 2D* 4). Hence, there was a profound abnormality of *Scn1a*$^{+/-}$ DG circuit function at the young adult timepoint, with a markedly higher proportion of GCs activated by PP input, as well as an increase in the number of action potentials fired in those GCs, versus age-matched wild-type controls, across stimulation paradigms and a broad range of intensities.

## Mechanisms of dentate gyrus circuit dysfunction in young adult *Scn1a+/-* mice

Increased PP activation of GCs could be due to increased intrinsic GC excitability, dysfunction of feedforward inhibition, or increased synaptic excitatory drive onto GCs. To investigate the mechanism of DG circuit dysfunction in young adult *Scn1a*$^{+/-}$ mice, we first assessed the intrinsic excitability of GCs in *Scn1a*$^{+/-}$ mice versus controls. Whole-cell current-clamp recordings of GCs from *Scn1a*$^{+/-}$ mice and age-matched wild-type littermate controls (at both the early postnatal and young adult timepoints) showed no differences across a range of measures of intrinsic excitability, properties of individual action potentials (APs), and repetitive AP firing (*Figure 2—figure supplement 4*; *Table 1*).

Single-cell electrophysiology data from acute brain slices and acutely dissociated neurons prepared from various *Scn1a*$^{+/-}$ mouse lines from multiple laboratories has repeatedly identified interneuron dysfunction (*Cheah et al., 2012*; *Dutton et al., 2017*; *Ogiwara et al., 2007*; *Richards et al., 2018*; *Rubinstein et al., 2015*; *Tai et al., 2014*), and in particular dysfunction of parvalbumin-expressing fast-spiking GABAergic interneurons (PV-INs) (*Dutton et al., 2013*; *Rubinstein et al., 2015*; *Tai et al., 2014*). Therefore, we next tested the intrinsic properties of DG PV-INs in *Scn1a*$^{+/-}$ mice versus wild-type littermate controls based on the involvement of these cells in DS pathogenesis as well as the fact that the GC response to PP input is known to be powerfully regulated by feedforward inhibition mediated by DG PV-INs (*Ewell and Jones, 2010*; *Lee et al., 2016*).

We prepared acute brain slices from early postnatal and young adult *Scn1a*$^{+/-}$ mice and wild-type littermates expressing tdTomato (tdT) under *Pvalb*-specific Cre-dependent control, as described previously (*Favero et al., 2018*). PV-INs in DG were thus identified by endogenous tdT expression visualized with epifluorescence and characteristic location at the GCL:hilus border. Early postnatal *Scn1a*$^{+/-}$ PV-INs exhibited profoundly impaired firing in response to depolarizing current steps (*Figure 3A–C*). The findings at the young adult timepoint were more subtle: young adult *Scn1a*$^{+/-}$ PV INs fired normally at the onset of a depolarizing current step and reached identical maximal steady-state firing frequencies but exhibit gradual spike height accommodation that ultimately progresses to spike failure with prolonged and large-amplitude (~2.5-fold rheobase) current injections (*Figure 3D–F*). Comparing across timepoints, while PV-INs from young adult *Scn1a*$^{+/-}$ mice continue to exhibit a reduced maximal

**Table 1.** Properties of early postnatal and young adult DG GCs from *Scn1a*⁺/⁻ and wild-type mice. For each parameter, overall genotype significance is determined by two-way ANOVA, with genotype comparisons at each timepoint calculated using Tukey's multiple comparisons tests.

| Measurement | Early postnatal | | | Young adult | | | Genotype variation (ANOVA) |
| --- | --- | --- | --- | --- | --- | --- | --- |
| | Scn1a⁺/⁻ | WT | p-value | Scn1a⁺/⁻ | WT | p-value | p-Value |
| *n* cells (mice) | 19 (3) | 21 (3) | | 16 (7) | 13 (4) | | |
| Age (days) | 18 ± 2 | 19 ± 1 | 0.88 | 55 ± 6 | 58 ± 2 | 0.75 | |
| Vm (mV) | −78.0 ± 1.6 | −77.6 ± 1.7 | 0.99 | −78.1 ± 1.6 | −78.8 ± 2.1 | 0.99 | 0.95 |
| Rm (MΩ) | 379 ± 19 | 405 ± 43 | 0.89 | 237 ± 20 | 217 ± 19 | 0.97 | 0.89 |
| Time Constant | 8.3 ± 1.4 | 9.1 ± 1.2 | 0.96 | 7.2 ± 0.8 | 8.1 ± 1.5 | 0.96 | 0.51 |
| Rheobase (pA) | 65 ± 8 | 69 ± 8 | 0.99 | 134 ± 16 | 119 ± 19 | 0.86 | 0.66 |
| AP Threshold (mV) | −36.2 ± 1.7 | −34.1 ± 2.8 | 0.89 | −37.2 ± 1.5 | −36.8 ± 1.6 | 0.99 | 0.56 |
| AP Amplitude (mV) | 80.9 ± 2.4 | 84.2 ± 1.8 | 0.64 | 81.4 ± 1.6 | 83.8 ± 2.5 | 0.88 | 0.19 |
| AP Peak (mV) | 44.7 ± 1.4 | 46.1 ± 2.2 | 0.94 | 44.2 ± 1.4 | 47.0 ± 2.1 | 0.77 | 0.27 |
| AP Rise Time (ms) | 0.54 ± 0.02 | 0.53 ± 0.03 | 0.97 | 0.57 ± 0.02 | 0.59 ± 0.02 | 0.94 | 0.87 |
| AP Halfwidth (ms) | 0.76 ± 0.02 | 0.80 ± 0.04 | 0.69 | 0.77 ± 0.03 | 0.86 ± 0.04 | 0.34 | 0.05 |
| AHP Amplitude (mV) | 15.2 ± 1.1 | 15.7 ± 0.7 | 0.98 | 13.9 ± 1.0 | 13.8 ± 1.1 | > 0.99 | 0.81 |
| AHP time (ms) | 3.08 ± 0.34 | 2.99 ± 0.27 | 0.99 | 1.91 ± 0.14 | 2.16 ± 0.09 | 0.93 | 0.09 |
| Sag (percent) | 3.2 ± 0.3 | 4.0 ± 0.5 | 0.71 | 3.5 ± 0.6 | 3.8 ± 0.9 | 0.98 | 0.32 |
| Max instantaneous (Hz) | 140 ± 10 | 160 ± 8 | 0.44 | 184 ± 13 | 161 ± 8 | 0.49 | 0.91 |
| Max steady-state (Hz) | 57 ± 3 | 65 ± 4 | 0.41 | 66 ± 4 | 63 ± 6 | 0.97 | 0.50 |

The online version of this article includes the following source data for table 1:

**Source data 1.** Data summary for all DG GCs electrophysiological data.

instantaneous firing frequency relative to age-matched wild-type (*Figure 3G*), there was normalization of maximal steady-state firing frequency as well as across multiple other metrics of intrinsic excitability and properties of individual action potentials (*Figure 3H–J*; *Table 2*).

Given near-normalization of PV-IN firing properties at the young adult timepoint, we reasoned that PV-IN dysfunction was unlikely to underlie the larger circuit deficit, which worsens, rather than improves, across development (*Figure 2*). To further confirm that the subtle deficits seen in individual PV-IN firing properties (*Figure 2*) were not responsible for the GC hyperactivation observed in young adult *Scn1a*⁺/⁻ mice, we performed an additional set of experiments using Hm1a, a peptide toxin that acts as an Nav1.1-specific activator (*Osteen et al., 2016*) and has been shown to correct the abnormalities seen in PV-INs in *Scn1a*⁺/⁻ mice (*Goff and Goldberg, 2019*). We found that bath-application of Hm1a to brain slices prepared from *Scn1a*⁺/⁻ mice corrected the PV-IN deficits apparent in response to large and prolonged current injections, whereas Hm1a had no impact on firing of PV-INs from wild-type mice, consistent with prior literature (*Richards et al., 2018*; *Figure 4A–D*). However, Hm1a had no effect on the large-scale evoked activation of GCs – as measured via 2 P imaging – in either genotype (*Figure 4E–H*). This result suggests that the subtle identified deficits in *Scn1a*⁺/⁻ PV-IN spike generation and impairment in repetitive firing does not underlie the observed large-scale circuit impairment at the young adult timepoint.

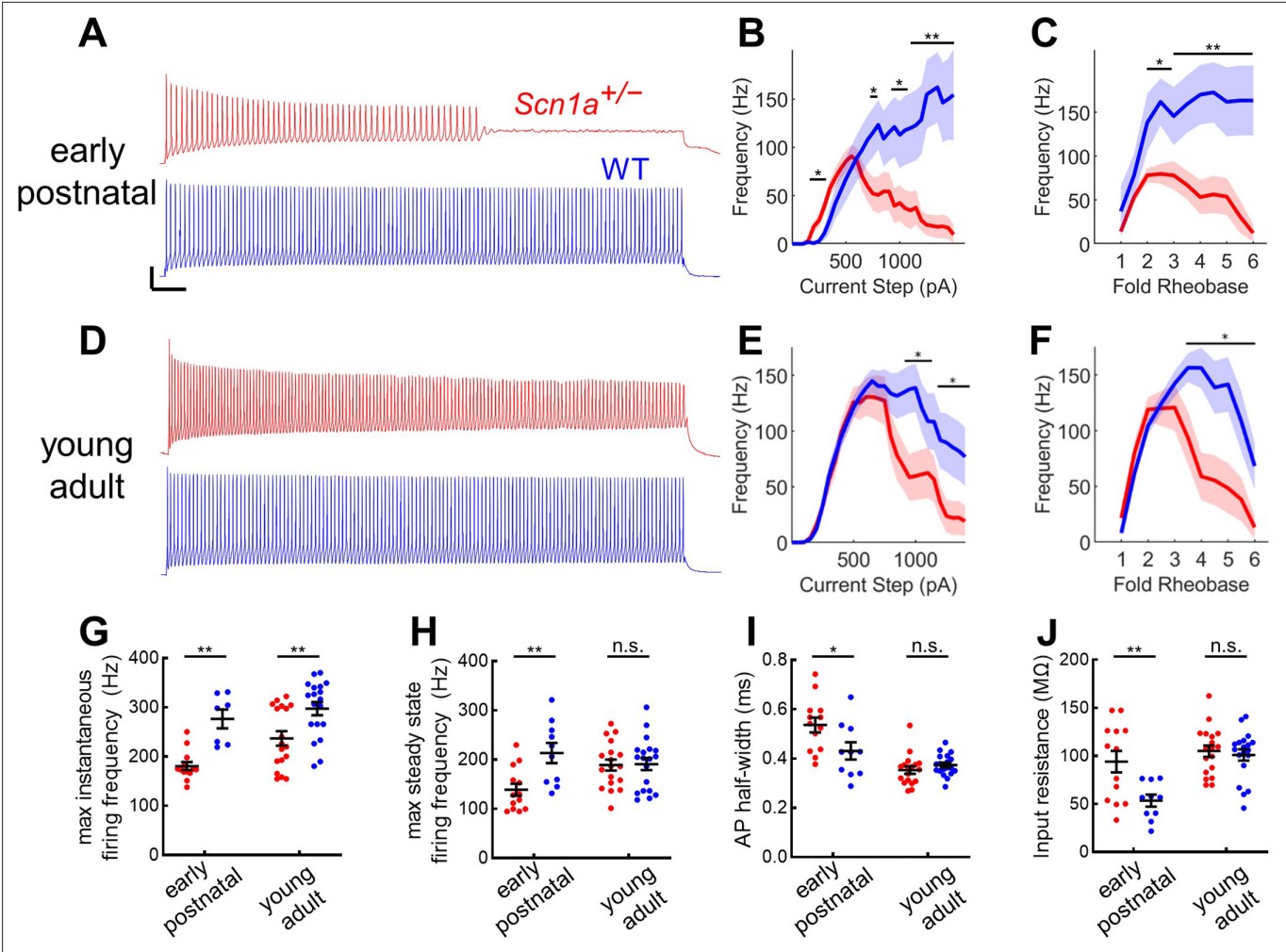

**Figure 3.** Profound impairment of spike generation in DG PV-INs from early postnatal *Scn1a*<sup>+/-</sup> mice, with partial normalization by young adulthood. (**A**) Example current clamp recordings of DG PV-INs from WT (*blue*) and *Scn1a*<sup>+/-</sup> (*red*) mice at the juvenile timepoint, demonstrating early spike failure in *Scn1a*<sup>+/-</sup> PV-INs. Scale bar 20 mV / 100ms. For juvenile GC PV-INs, (**B**) current/frequency (I-f) plot and (**C**) I-F plot with current normalized to Rheobase for each cell. (**D**) Example current clamp recordings from WT (*blue*) and *Scn1a*<sup>+/-</sup> (*red*) DG PV-INs at the young adult timepoint, demonstrating progressive spike-height accommodation in *Scn1a*<sup>+/-</sup> PV-INs in response to a prolonged depolarizing current step. Scale bar as in **A**. For young adult GC PV-INs, (**E**) current/frequency (I-F) plot and (**F**) I-F plot with current normalized to Rheobase for each cell. *Scn1a*<sup>+/-</sup> PV-INs have significantly lower instantaneous firing frequency at both timepoints (**G**) but the steady state firing frequency normalizes by the young adult timepoint (**H**). *Scn1a*<sup>+/-</sup> PV-INs display larger input resistance (**I**) and action potential half-width (**J**) at the early postnatal timepoint only. For **B–F**, line and shaded areas represent mean and SEM, and bars indicate significance calculated using one-way ANOVA and post-hoc tests with Bonferroni correction. For **G–J**, significance is determined by Tukey's multiple comparisons test. *, $p < 0.05$; **, $p < 0.01$; ***, $p < 0.001$. *n* and mouse ages are as per *Table 2*. See also *Table 2—source data 1*.

Since impaired response of DG to entorhinal cortex input in *Scn1a*<sup>+/-</sup> mice could not be adequately attributed to aberrant intrinsic properties of either DG GCs or PV-INs, we next considered whether this finding may instead be due to alterations in the excitatory drive onto GCs and/or recruitment of disynaptic inhibition. We performed whole-cell voltage-clamp recordings of GCs from *Scn1a*<sup>+/-</sup> mice versus wild-type controls (P54-75) using a cesium-based internal solution to isolate evoked monosynaptic excitatory postsynaptic currents (EPSCs; recorded at –70 mV) and di-synaptic inhibitory postsynaptic currents (IPSCs; recorded at +10 mV) (*Figure 5A*). We first quantified the minimal PP stimulation (tested in ascending 25 µA increments) required to evoke an EPSC and IPSC for each cell (*Figure 5— figure supplement 1*) and did not find a significant difference between genotypes (EPSC: 42.5 ± 5.3 µA in *Scn1a*<sup>+/-</sup> versus 49.7 ± 6.6 WT, p = 0.41; IPSC: 27.1 ± 2.4 *Scn1a*<sup>+/-</sup> versus 36.6 ± 3.9 WT, p = 0.06).

We next calculated the ratio of the maximal evoked EPSC/IPSC amplitude (*Figure 5B*). We tested the null hypothesis that all data could be fit by one curve (one-site binding curve), and found that,

**Table 2.** Properties of early postnatal and young adult DG PV-INs from *Scn1a*$^{+/-}$ and wild-type mice. For each parameter, overall genotype significance is determined by two-way ANOVA, with genotype comparisons at each timepoint calculated using Tukey's multiple comparisons tests.

| Measurement | Early postnatal | | | Young adult | | | Genotype variation (ANOVA) |
| --- | --- | --- | --- | --- | --- | --- | --- |
| | Scn1a$^{+/-}$ | WT | p-value | Scn1a$^{+/-}$ | WT | p-value | p-Value |
| *n* cells (mice) | 13 (5) | 10 (5) | | 18 (5) | 19 (4) | | |
| Age (days) | 19 ± 1 | 19 ± 1 | 0.72 | 69 ± 3 | 66 ± 2 | 0.56 | |
| Vm (mV) | −57.9 ± 1.7 | −61.5 ± 2.3 | 0.65 | −57.6 ± 1.8 | −57.2 ± 1.8 | 0.99 | 0.42 |
| Rm (MΩ) | 94 ± 11 | 53 ± 6 | 0.007 (**) | 105 ± 6 | 101 ± 6 | 0.97 | 0.005 (**) |
| Time Constant | 5.3 ± 0.8 | 4.7 ± 1.0 | 0.91 | 6.4 ± 0.3 | 5.1 ± 0.3 | 0.26 | 0.10 |
| Rheobase (pA) | 555 ± 96 | 802 ± 120 | 0.10 | 489 ± 35 | 463 ± 34 | 0.99 | 0.10 |
| AP Threshold (mV) | −35.7 ± 3.4 | −45.8 ± 3.1 | 0.04 (*) | −50.4 ± 1.4 | −49.0 ± 1.6 | 0.96 | 0.07 |
| AP Amplitude (mV) | 64.2 ± 2.1 | 61.8 ± 4.2 | 0.92 | 75.1 ± 1.7 | 76.9 ± 1.9 | 0.93 | 0.91 |
| AP Peak (mV) | 28.5 ± 3.4 | 16.1 ± 4.0 | 0.008 (**) | 24.7 ± 31.3 | 27.9 ± 1.3 | 0.69 | 0.05 |
| AP Rise Time (ms) | 0.63 ± 0.15 | 0.50 ± 0.07 | 0.61 | 0.67 ± 0.08 | 0.50 ± 0.05 | 0.41 | 0.06 |
| AP Halfwidth (ms) | 0.54 ± 0.03 | 0.43 ± 0.04 | 0.01 (*) | 0.35 ± 0.01 | 0.37 ± 0.01 | 0.86 | 0.048 (*) |
| AHP Amplitude (mV) | 10.7 ± 1.5 | 7.2 ± 3.5 | 0.58 | 11.4 ± 1.1 | 9.7 ± 1.2 | 0.85 | 0.14 |
| AHP time (ms) | 1.60 ± 0.11 | 1.12 ± 0.10 | 0.001 (**) | 0.98 ± 0.05 | 0.93 ± 0.03 | 0.92 | 0.0008 (***) |
| Sag (percent) | 11.1 ± 1.7 | 26.4 ± 6.6 | 0.007 (**) | 9.0 ± 1.1 | 17.3 ± 2.3 | 0.10 | 0.0001 (***) |
| Max instantaneous (Hz) | 181 ± 8 | 277 ± 19 | 0.002 (**) | 237 ± 15 | 297 ± 13 | 0.006 (**) | < 0.0001 (****) |
| Max steady-state (Hz) | 139 ± 12 | 213 ± 20 | 0.006 (**) | 189 ± 11 | 191 ± 12 | 0.99 | 0.008 (**) |

The online version of this article includes the following source data for table 2:

**Source data 1.** Data summary for all DG PV-IN electrophysiological data.

instead, the fits of the two genotypes were significantly different (p < 0.0001), with a markedly larger EPSC/IPSC ratio in *Scn1a*$^{+/-}$ GCs. The data were further analyzed using a mixed-effects model with post-hoc analysis with multiple comparisons, with significant differences found between genotypes in the two strongest stimulation conditions (400 µA: 1.8 ± 0.4 *Scn1a*$^{+/-}$ versus 1.0 ± 0.2 WT, p = 0.03; 500 µA: 1.9 ± 0.5 *Scn1a*$^{+/-}$ versus 1.0 ± 0.1 WT, p = 0.009).

To understand whether this result was driven by increased excitation and/or decreased inhibition, we examined the raw EPSC and IPSC magnitudes. We found significantly larger EPSC magnitudes (*Figure 5C*) in *Scn1a*$^{+/-}$ versus wild-type GCs (p < 0.0001, analysis of curve fits), with over two-fold higher EPSC magnitudes in *Scn1a*$^{+/-}$ (e.g. at 400 µA: 825 ± 159 vs 366 ± 88 pA, p < 0.0001, using mixed-effects analysis with post-hoc correction for multiple comparisons). Overall the IPSC magnitude also differed between genotypes (p = 0.01, analysis of curve fits), with IPSC amplitude trending larger (not smaller, as would contribute to a larger E/I ratio) in *Scn1a*$^{+/-}$ versus wild-type GCs, although the difference was not enough to reach significance at any individual data point on post-hoc analysis (*Figure 5D*).

Overall, these data suggest that hyper-activation of GCs in young adult *Scn1a*$^{+/-}$ mice results from increased excitatory drive to DG rather than impaired inhibition. If anything, disynaptic inhibition – mediated largely by PP-driven recruitment of PV-INs – is normal or enhanced, although unable to

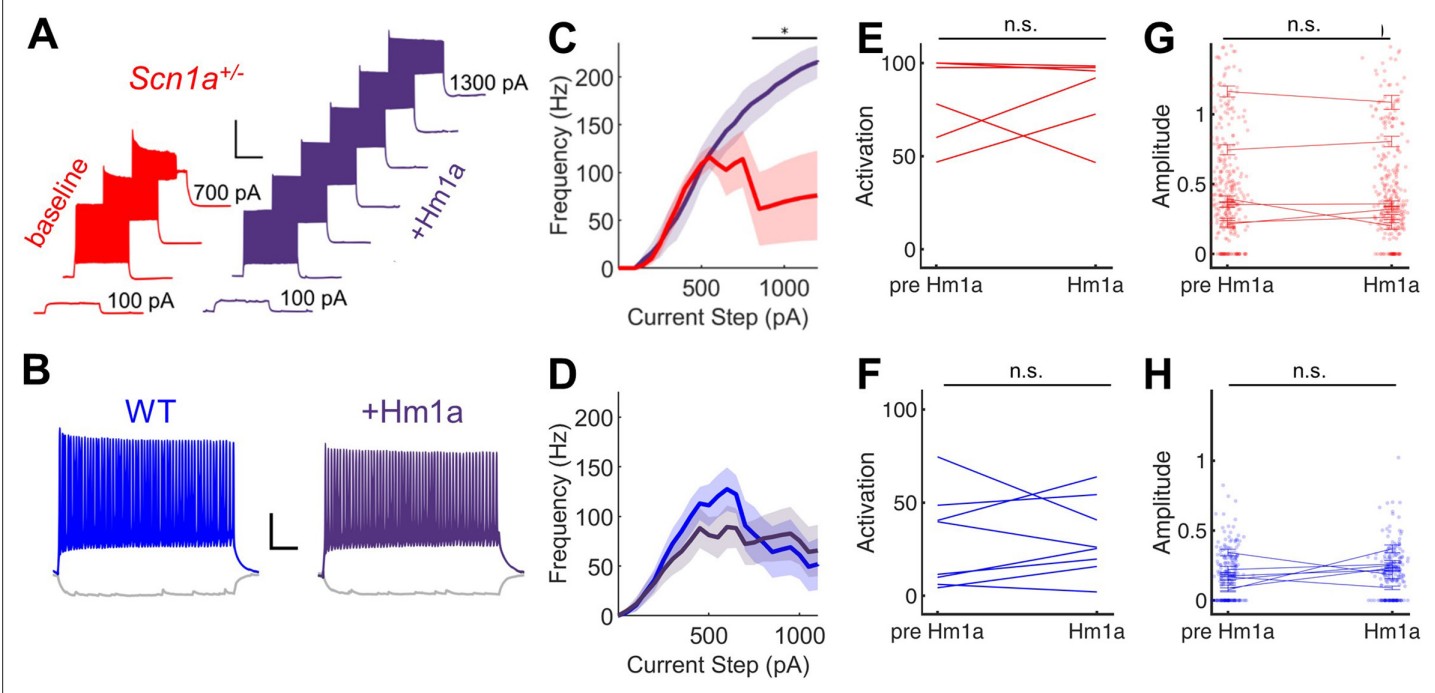

**Figure 4.** Hm1a enhances fast-spiking discharge properties in DG PV-INs from *Scn1a*⁺/⁻ (but not wild-type) mice but has no effect on evoked GC activation. (**A**) Example trace from one *Scn1a*⁺/⁻ PV-IN showing responses to depolarizing current steps at baseline (*red*) or in the presence of Hm1a (*purple*). Hm1a decreased spike height accommodation and prevented AP failures with larger current injections. Scale bar 300 ms / 50 mV. Note that single APs cannot be visualized due to condensed timescale and high frequency firing. (**B**) Example trace from one wild-type PV-IN at baseline (*blue*) or in the presence of Hm1a (*purple*). Scale bar 100 ms / 20 mV. Current/frequency (*I*-F) plots at baseline and in the presence of Hm1a for *Scn1a*⁺/⁻ PV-INs (**C**) and wild-type PV-INs (**D**), showing significant enhancement of firing in *Scn1a*⁺/⁻ PV-INs only. Line and shaded areas represent mean and SEM. Bars indicate significance calculated using one-way ANOVA and post-hoc tests with Bonferroni correction: *, p < 0.05. For both genotypes, bath-application of Hm1a does not alter the perforant path-evoked proportional activation of GCs (**E–F**) or the amplitude of the calcium signal within those activated cells (**G–H**). *n* for *Scn1a*⁺/⁻ and WT, respectively: 5 and 9 (cells), 3 and 4 (mice) for electrophysiology; 335 and 465 (cells), 6 and 8 (fields), 2 and 3 (mice) for 2 P imaging.

The online version of this article includes the following source data for figure 4:

**Source data 1.** Quantified Hm1a imaging data, reported by cell.

**Source data 2.** Quantified Hm1a imaging data, reported by field.

balance the greatly enhanced excitatory drive at the circuit level. To corroborate this finding, we further analyzed the calcium imaging data obtained in the presence of a saturating concentration of PTX. We postulated that, if decreased GABA_A receptor-mediated inhibition was the cause of GC hyperactivation in *Scn1a*⁺/⁻ mice, then PTX should eliminate the difference between genotypes seen in **Figure 2**. However, we found that the genotype difference persisted in the presence of PTX, with mixed model analysis of d*F*/*F*0 magnitudes being larger for *Scn1a*⁺/⁻ (3.64 ± 0.07) versus wild-type (2.21 ± 0.10; p < 0.001) in the presence of PTX, further consistent with the conclusion that there is increased PP-mediated excitatory drive onto DG granule cells in *Scn1a*⁺/⁻ mice during the chronic phase of pathology (**Figure 5E**).

This increase in evoked EPSC could result from a pre- and/or post-synaptic mechanism. To investigate this, we first quantified responses elicited by brief puffs of glutamate (1 mM) onto GC dendrites, delivered via a pipette placed in a stereotyped position in the DG molecular layer (**Figure 5F**). We saw no significant genotype difference in the resulting glutamate-evoked inward currents at any tested stimulation duration (**Figure 5G**; **Figure 5—figure supplement 1**). We next measured the GC response to repetitive PP stimulation and quantified the peak amplitude of the 2nd / 1st EPSC (i.e. the paired-pulse ratio; PPR), as well as the peak amplitude of the 4th / 1st EPSC, as an indicator of release probability. We found significantly lower ratios in *Scn1a*⁺/⁻ GCs (**Figure 5H–I**; **Figure 5—figure supplement 1**), suggesting that the release probability may be higher at the *Scn1a*⁺/⁻ PP-to-GC synapse.

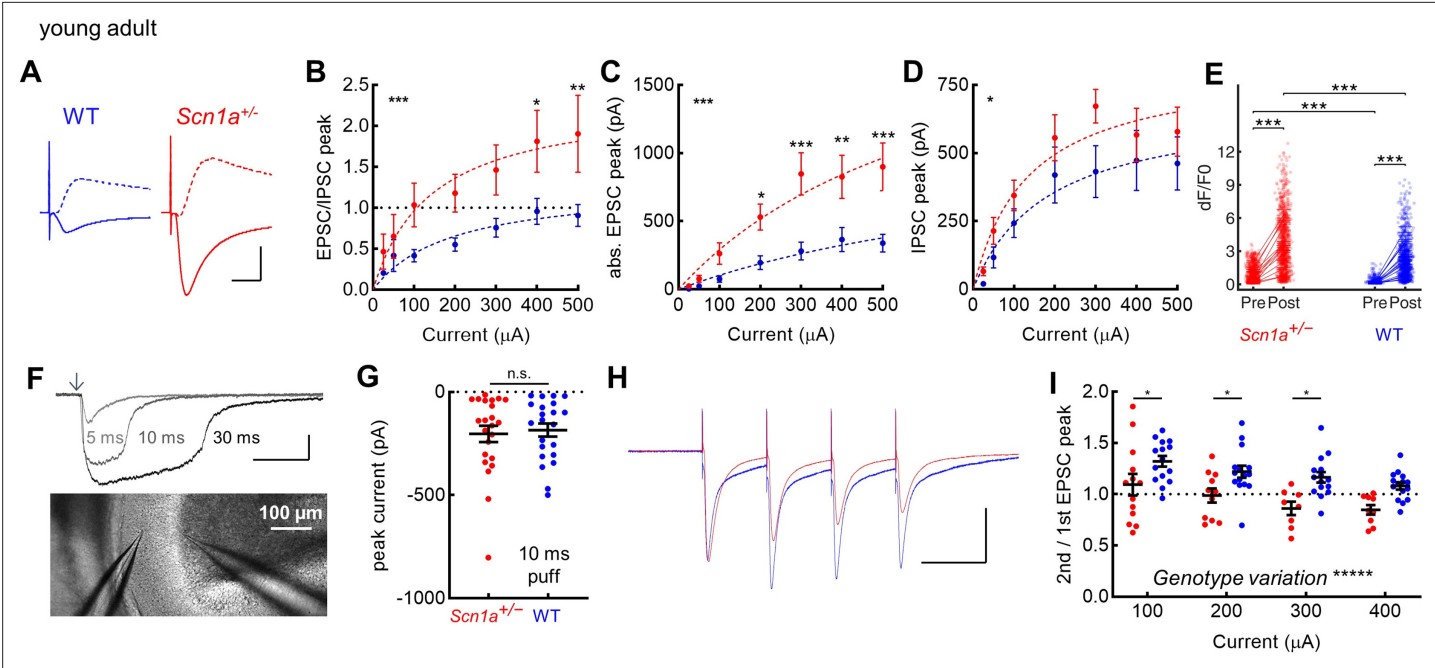

**Figure 5.** Selective increase in PP-evoked excitation of young adult *Scn1a*+/- dentate gyrus granule cells. Whole cell patch clamp recordings were performed in GCs from *Scn1a*+/- and wild-type mice (P54-75). (**A**) Representative traces from wild-type (*blue*) and *Scn1a*+/- (*red*) GCs showing evoked monosynaptic EPSCs (recorded at –70 mV; *solid*) and di-synaptic IPSCs (recorded at +10 mV; *dashed*) in response to 300 µA PP stimulation. Scale bar, 10 ms / 500 pA. (**B**) EPSC / IPSC magnitude was calculated in response to 100–500 µA PP input. Data were fit with a one site binding curve and compared with an extra sum-of-squares F test (p < 0.0001). Mixed-effects analysis with multiple comparisons were used for comparisons at each point (p = 0.03 for 400 µA and p = 0.009 for 500 µA). (**C**) Raw evoked ESPC magnitude was significantly larger in *Scn1a*+/- GCs (p < 0.0001; extra sum-of-squares F test comparison of curve fits). Mixed-effects analysis with multiple comparisons were used for comparisons at each point (p = 0.01 for 200 µA, 0.004 for 400 µA, and <0.001 for 300 and 500 µA). (**D**) Raw evoked disynaptic IPSC magnitude was also higher overall in *Scn1a*+/- GCs (p = 0.01, extra sum-of-squares F test comparison of curve fits), although significance was not reached at any individual data point. *n* for the experiments in B-D was, *Scn1a*+/- and WT, respectively: 13 and 16 (cells), 4 and 4 (mice). Stars in the upper left indicate significance of overall curve fits while stars above individual data points indicate post-hoc significance at each point. (**E**) Magnitude of evoked GC responses as quantified by calcium imaging (d*F*/*F*0) in the presence of 100 µM picrotoxin, with persistence of significantly larger responses measured in *Scn1a*+/- GCs in the setting of GABA_A receptor blockade (p < 0.001; mixed model analysis). *n* for *Scn1a*+/- and WT, respectively: 1073 and 1109 (cells), 14 and 16 (both fields and slices), 7 and 6 (mice). (**F**) Puffed glutamate (1 mM) was applied to the mid-molecular layer, with pulses of 5, 10, or 30 ms duration, while resulting currents were recorded from GCs. Scale bar 100 pA / 100ms. Arrow indicates delivery of pressure pulse to the glutamate-containing pipette. (**G**) Currents evoked by 10 ms glutamate puffs. *n* for *Scn1a*+/- and WT, respectively: 4 and 4 (mice), 23 and 22 (cells). (**H**) Representative scaled traces from wild-type and *Scn1a*+/- GCs showing EPSCs in response to 300 µA PP stimulation delivered at 20 Hz. Scale bar, 400 pA (*Scn1a*+/-) / 300 pA (wild-type) / 40ms. The decay of the EPSC (from 300 µA stimulation) was fit with a single exponential that was not different by genotype: 7.2 ± 0.6ms for *Scn1a*+/- (n = 8) and 8.1 ± 0.7ms for WT (n = 15; p = 0.44 vs. WT via unpaired t-test). (**I**) EPSC PPR comparing 2nd / 1st PP stimuli. There was a highly significant overall effect of genotype (p < 0.0001; two-way ANOVA) as well as significant genotype differences with individual comparisons (Sidak's multiple comparisons test: 100 µA, p = 0.04; 200 µA, p = 0.04; 300 µA, p = 0.01). *n* for *Scn1a*+/- and WT, respectively: 14 and 16 (cells), 4 and 4 (mice).

The online version of this article includes the following source data and figure supplement(s) for figure 5:

**Source data 1.** Young adult evoked response data.

**Source data 2.** Early postnatal evoked response data.

**Figure supplement 1.** Additional metrics of evoked post-synaptic responses in young adult dentate gyrus granule cells.

**Figure supplement 2.** Evoked excitation is not increased in early postnatal *Scn1a*+/- dentate gyrus granule cells.

In contrast, parallel experiments at the early postnatal timepoint revealed no genotype difference in either glutamate puff-evoked currents or in the dynamics of perforant path-evoked EPSCs (*Figure 5—figure supplement 2*). However, the small but statistically significant genotype difference in d*F*/*F*0 (*Figure 2D* 1) persisted in the presence of PTX (*Figure 5—figure supplement 2A*) and is thus also independent of synaptic inhibition.

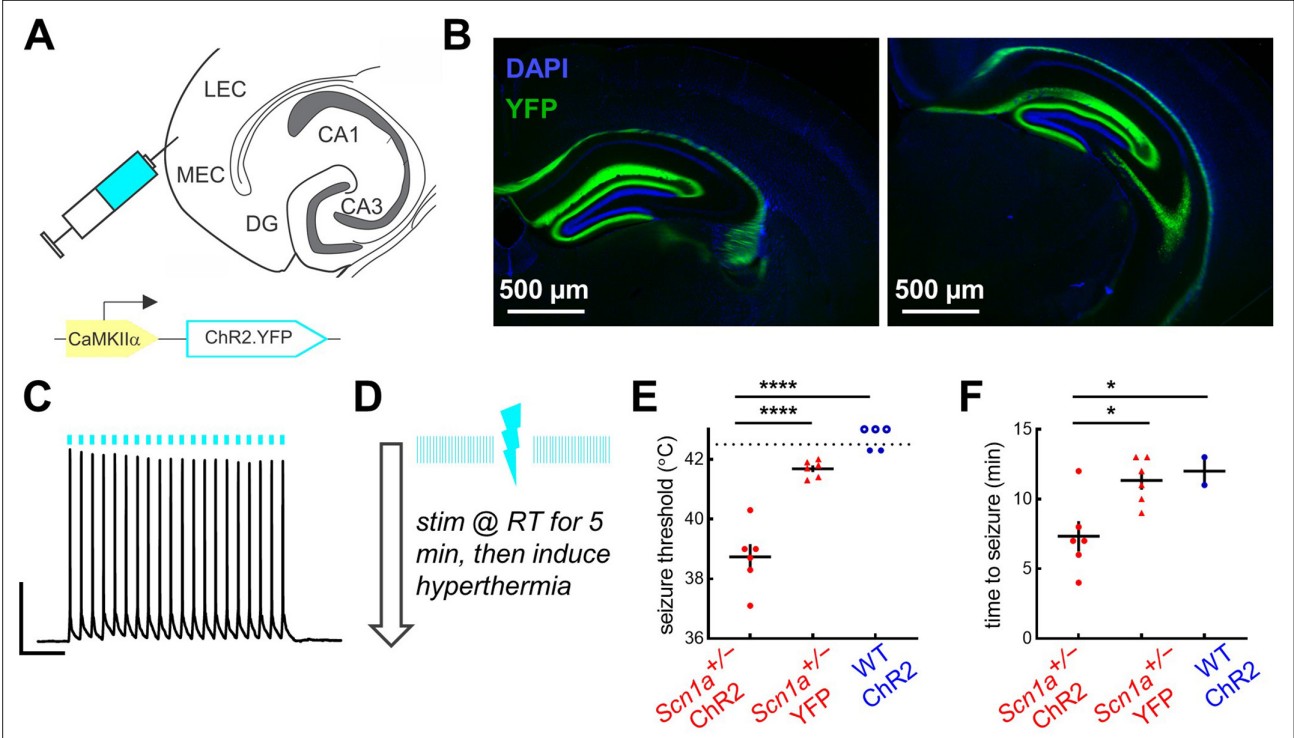

**Figure 6.** Activation of entorhinal cortex lowers temperature threshold for seizure induction in *Scn1a*$^{+/-}$ mice. (**A**) *Scn1a*$^{+/-}$ and wild-type mice were injected with AAV9-CaMKIIα-ChR2-YFP into entorhinal cortex; a subset of *Scn1a*$^{+/-}$ mice were instead injected with the control virus AAV9-CaMKIIα-YFP. (**B**) Viral expression (*green*) was seen in the PP projection to dorsal (*left*) and ventral (*right*) hippocampus (as delineated by DAPI; *blue*). (**C**) Whole-cell current-clamp trace demonstrating light-evoked 20 Hz action potentials in a ChR2-expressing neuron in entorhinal cortex (470 nm, *blue lines*). Scale bar: 200 ms / 40 mV. (**D**) Pulsed photostimulation is delivered to EC at 20 Hz, 5ms pulse-width, for 5 s on / 5 s off. Mice are stimulated initially at room temperature for up to 5 min, then while subjected to hyperthermia again until 42.5 °C, or until a behavioral seizure is observed. (**E**) Seizure threshold for *Scn1a*$^{+/-}$ mice expressing ChR2 (38.7°C ± 0.5°C; *n* = 6 mice; *red circles*) was significantly lower than that for *Scn1a*$^{+/-}$ mice expressing YFP control virus (41.7°C ± 0.1°C; *n* = 6 mice; *red triangles*), or wild-type mice expressing ChR2 (*n* = 5 mice; *blue circles*). Note that some wild-type mice had no seizure prior to 42.5 °C (*open circles*). p < 0.0001 for both two-way comparisons (Sidak's multiple comparisons tests, assuming a threshold of 42.5 °C for the wild-type mice that had no seizure). (**F**) Photostimulation duration prior to seizure onset was also significantly lower for the *Scn1a*$^{+/-}$-ChR2 group (p = 0.02 versus *Scn1a*$^{+/-}$-YPF and 0.04 versus wild-type-ChR2; Sidak's multiple comparisons tests).

The online version of this article includes the following source data for figure 6:

**Source data 1.** Seizure threshold and time to seizure data.

## Optogenetic activation of entorhinal cortex readily triggers seizures in *Scn1a+/-* mice

The DG controls hippocampal signal propagation from entorhinal cortex to CA3; dysfunction of this control point is hypothesized to be involved in temporal lobe epileptogenesis and seizure generation in chronic temporal lobe epilepsy (*Behr et al., 1998*; *Dengler et al., 2017*; *Lu et al., 2016*). Having demonstrated that corticohippocampal circuit function is impaired in *Scn1a*$^{+/-}$ mice (*Figure 2*), we next moved in vivo to test the implications of this circuit dysregulation on ictogenesis. We hypothesized that the hyperexcitable corticohippocampal circuit in *Scn1a*$^{+/-}$ mice would be vulnerable to seizures in response to entorhinal cortex activation.

We injected an AAV encoding ChR2 under control of the excitatory neuron-specific promoter, CaMKIIα, into the entorhinal cortex of *Scn1a*$^{+/-}$ and wild-type mice (*Figure 6A*) and verified that these mice had robust viral expression throughout the molecular layer of dorsal and ventral hippocampus (*Figure 6B*). A separate group of *Scn1a*$^{+/-}$ mice were injected with an AAV encoding a fluorophore alone, as a control for potential effects of heating due to photostimulation (*Owen et al., 2019*). We confirmed functional ChR2 expression within entorhinal cortex, with action potentials driven by photostimulation of ChR2-expressing neurons (*Figure 6C*).

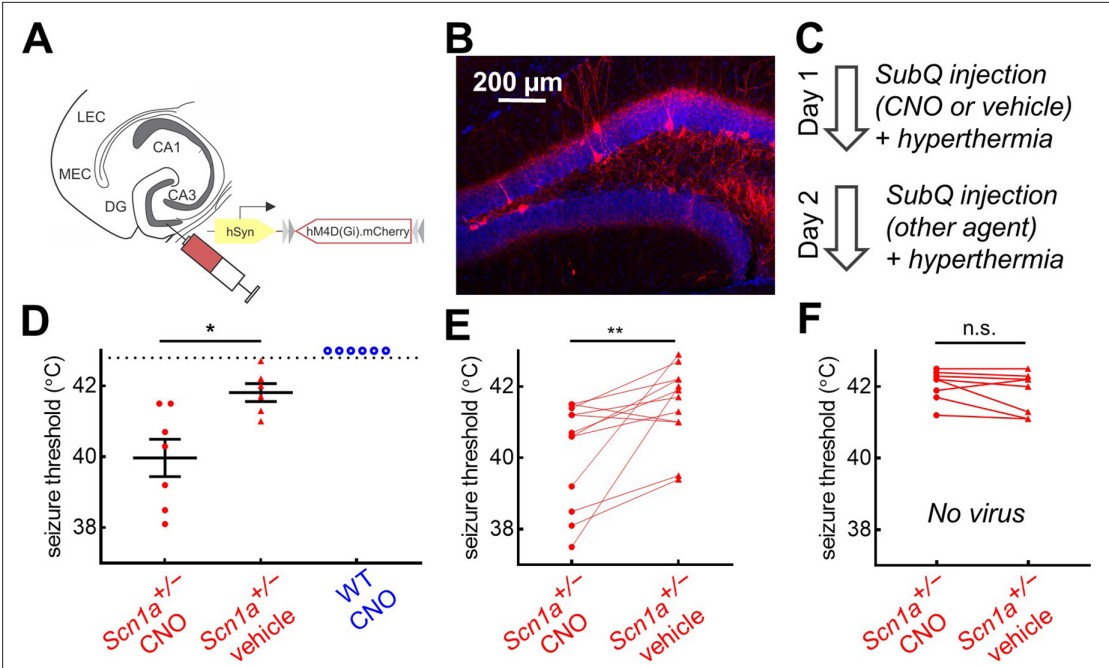

**Figure 7.** Chemogenetic inhibition of PV-INs lowers temperature threshold for seizure induction in *Scn1a*[+/-] mice. (**A**)*Scn1a*[+/-] and wild-type *Pvalb*[Cre] mice were injected with AAV9-hSyn-FLEX-hM4D(Gi)-mCherry into bilateral DG. (**B**) PV-IN expression (*red*) was sparse within the dentate gyrus (anatomically delineated by DAPI staining, *blue*). (**C**) On Day 1, mice were randomized to receive either CNO (*circle*) or vehicle (*triangle*) injection prior to hyperthermic seizure induction; mice received the other agent on Day 2. (**D**) The seizure threshold for mice with CNO injection (i.e. inhibition of dentate PV-INs) was significantly lower than vehicle (p = 0.01; unpaired t test); no wild-type mice had evoked seizures after CNO injection (*open circles*). (**E**) Data pooled across the two days revealed a significantly lower seizure threshold with CNO injection (p = 0.009; paired t test). (**F**) In a separate control cohort of mice not injected with hM4D(Gi), there was no effect of CNO on seizure threshold. *n* = 12 *Scn1a*[+/-] mice with hM4D(Gi), 6 wild-type mice with hM4D(Gi), and 8 *Scn1a*[+/-] mice with no viral injection.

The online version of this article includes the following source data for figure 7:

**Source data 1.** Seizure threshold data.

It is well established that *Scn1a*[+/-] mice exhibit not only spontaneous but also temperature-sensitive seizures (*Oakley et al., 2009*). We took advantage of this phenotype to test whether activation of entorhinal cortex could also shift the temperature threshold for seizure generation in *Scn1a*[+/-] mice. We optogenetically stimulated entorhinal cortex in vivo, initially at room temperature for 5 min, and then (if no seizure was evoked), while subjecting the mice to a ramp of increased core body temperature up to 42.5 °C (*Figure 6D*). Since rhythmic stimulation of excitatory neurons can evoke seizures even in wild-type mice (*Khoshkhoo et al., 2017*), we titrated our stimulation parameters (5 ms pulse width at 20 Hz = 10% duty cycle; 5 s on / 5 s off; similar to previously published protocol *Leung et al., 2018*) such that the wild-type ChR2 mice had either no seizures, or seizures only at/near maximal temperature elevation (42.5 °C). Under these same conditions, the *Scn1a*[+/-]-ChR2 mice exhibited behavioral seizures at a significantly lower threshold temperature compared to both control cohorts (*Figure 6E*). A statistically significant difference was also seen in time to seizure onset during stimulation (*Figure 6F*), although noting that any cohort differences in autonomic regulation and thus heating rate could be confounding (*Sahai et al., 2021*).

### Inhibition of dentate gyrus PV-INs lowers the temperature threshold for seizure induction in *Scn1a+/-* mice

If corticohippocampal circuit hyperexcitability promotes susceptibility to hyperthermic seizures and is not due to DG PV-IN dysfunction, we hypothesized that inhibition of DG PV-INs would exacerbate the hyperthermic seizure phenotype of *Scn1a*[+/-] mice. To test this, we injected an AAV encoding the inhibitory Designer Receptor Exclusively Activated by Designer Drugs (DREADD; AAV8.hSyn. FLEX.hMD4(Gi).mCh; *Krashes et al., 2011*) throughout the bilateral dentate gyri of *Pvalb*[Cre] mice

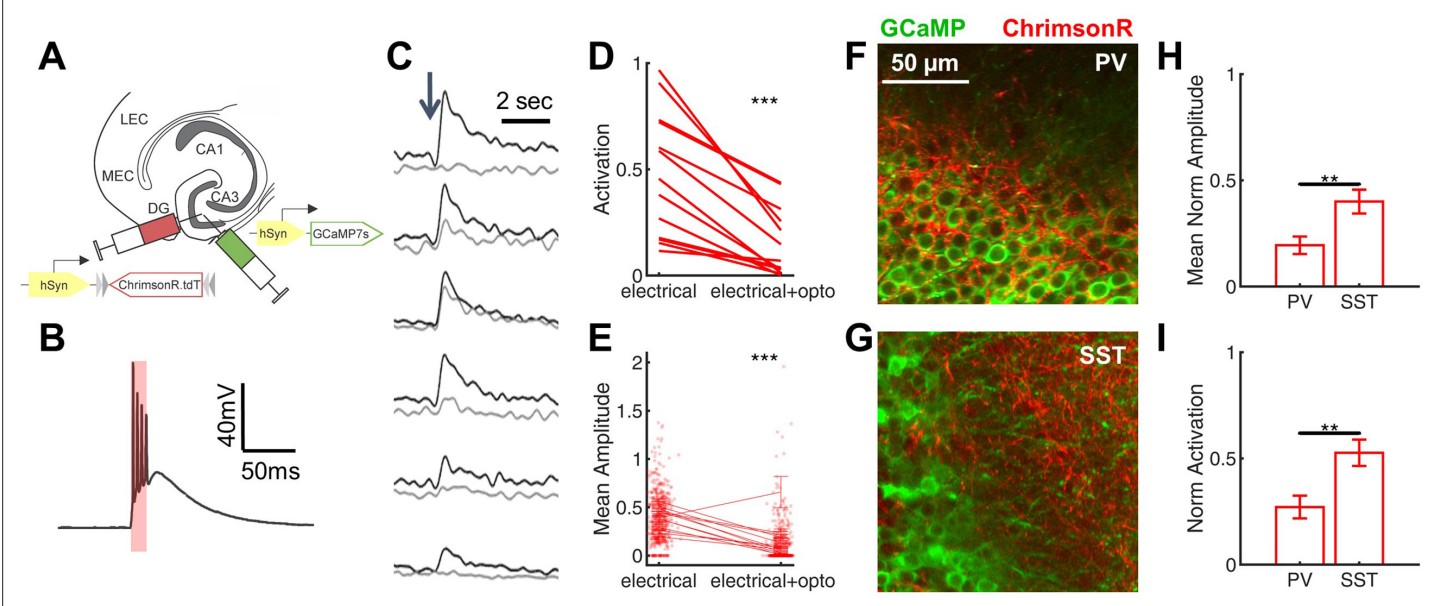

**Figure 8.** Rescue of corticohippocampal hyperexcitability via optogenetic activation of PV interneurons. (**A**) *Scn1a*+/-.*Pvalb*Cre mice were injected with a mix of AAV9-hSyn-GCaMP7s and AAV9-hSyn-FLEX-ChrimsonR-tdT into DG. (**B**) Whole-cell current-clamp trace demonstrating light-evoked action potentials in a ChrimsonR-expressing PV-IN (660 nm, *red bar*). (**C**) Example calcium transients in GCs with electrical PP stimulation (single pulse, 500 µA, *arrow*), either alone (*black*) or coupled with optogenetic PV-IN activation (10ms light pulse; *gray*). Proportional GC activation (**D**) and magnitude of GC responses (**E**) to PP stimulation (one pulse, 500 µA) are both significantly decreased by concurrent optogenetic PV-IN activation. Each line represents one imaging field, subject to PP stimulation alone versus PP stimulation with optogenetic PV-IN activation (p < 0.001; paired t-test). Expression in PV-INs (**F**) results in perisomatic ChrimsonR fibers (*red*) surrounding GCaMP-expressing GC cells (*green*) within the granule cell layer, whereas expression in SST-INs (**G**) results in ChrimsonR expression primarily seen within the molecular layer. Optogenetic activation of DG PV-INs is more effective that activation of DG SST-INs at decreasing PP stimulation-evoked activity in GCs, with a lower proportional activation (**G**) and d*F/F*0 amplitude (**H**), normalized to the response to PP stimulation alone in the same cells and imaging fields. Significance (p < 0.01) determined using mixed model analysis. *n* for *Pvalb*Cre and *Sst*Cre, respectively: 1055 and 663 (cells), 13 and 15 (fields), 6 and 3 (mice).

The online version of this article includes the following source data and figure supplement(s) for figure 8:

**Source data 1.** Quantified imaging data with optogenetic activation of PV- and SST-INs, reported by cell.

**Source data 2.** Quantified imaging data with optogenetic activation of PV- and SST-INs, reported by field.

**Figure supplement 1.** *Pvalb*Cre and *Sst*Cre lines have high specificity in dentate gyrus.

(*Figure 7A*). We verified virus expression within DG (*Figure 7B*) and performed acute whole-cell recordings in slice to confirm that bath-application of the ligand clozapine-N-oxide (CNO) significantly hyperpolarized expressing neurons (change in resting potential –9.9 ± 2.8 mV, p = 0.02, paired t-test, n = 6 cells).

We tested the effect of PV-cell inhibition in vivo by comparing the temperature threshold for seizure induction of mice following subcutaneous injection of CNO versus vehicle control (*Figure 7C*). We found that *Scn1a*+/- mice injected with CNO had significantly decreased seizure threshold; wild-type mice did not exhibit behavioral seizures despite CNO injection (*Figure 7D*). We then performed a crossover the following day, such that mice that received vehicle on Day one received CNO on Day 2, and vice versa, which further demonstrated that inhibition of PV-INs lowered seizure threshold in *Scn1a*+/- mice (*Figure 7D*). A separate, control cohort of *Scn1a*+/- mice, with no viral DREADD expression, were used to confirm that CNO itself was not ictogenic (*Figure 7E*).

## Activation of dentate gyrus PV-INs decreases corticohippocampal circuit hyperexcitability in *Scn1a*+/- mice

Given that PV-INs are known to exert powerful feedforward inhibition in DG (*Ewell and Jones, 2010*; *Lee et al., 2016*), and that *Scn1a*+/- DG PV-INs have only subtle deficits of spike initiation at the young adult time point (*Figure 3*, *Table 2*), we hypothesized that targeted activation of DG PV-INs could correct the imbalance of evoked inputs onto GCs in *Scn1a*+/- mice. We injected a 1:1 mix of two AAVs

into the DG of $Scn1a^{+/-}.Pvalb^{Cre}$ mice (*Figure 8A*): (1) the calcium indicator GCaMP7s (AAV9.hSyn. GCaMP7s) as in *Figure 1*, and (2) the red-shifted excitatory opsin, ChrimsonR (*Klapoetke et al., 2014*) under Cre-dependent control (AAV9.hSyn.FLEX.ChrimsonR.tdT), allowing simultaneous imaging of DG and photo-activation of PV-INs.

After verifying functional opsin expression in PV-INs (*Figure 8B*), we used 2 P calcium imaging to record PP stimulation-evoked activation of GCs either with or without concurrent optogenetic activation of DG PV-INs. We found that optogenetic co-activation significantly reduced PP-evoked DG hyper-activation in $Scn1a^{+/-}$ mice (*Figure 8C–E*), decreasing the proportion of activated GCs (0.15 ± 0.04 with PV-IN activation versus 0.48 ± 0.08 without; p < 0.001, paired t-test) and the magnitude of evoked calcium transient (d*F*/F0) within that subset of activated cells (0.10 ± 0.03 with PP stimulation versus 0.38 ± 0.03 without; p < 0.001; paired t-test). As a negative control, we also delivered photo-stimulation without PP activation in a subset of imaging fields, and confirmed that, as expected, no GCs were directly activated (data not shown).

To test whether this rescue was specific to activation of PV-INs, we repeated the experiment with optogenetic activation of SST-INs (using $Scn1a^{+/-}.Sst^{Cre}$ mice). Both $Pvalb^{Cre}$ and $Sst^{Cre}$ lines demonstrated specificity via immunohistochemical confirmation (*Figure 8—figure supplement 1*) and via expected differences in localization of ChrimsonR-expressing fibers within dentate (*Figure 8F–G*), corresponding with known perisomatic (PV) versus dendritic (SST) inhibition of GCs (*Amaral et al., 2007*; *Houser, 2007*). We found that SST-IN activation did also decrease GC responses to PP stimulation (*Figure 8H–I*), but to a significantly lesser degree (GC proportion of cells activated decreased to 0.27 ± 0.05 of baseline with PV-IN versus 0.53 ± 0.06 of baseline with SST-IN photostimulation; GC calcium transient amplitudes decreased to 0.19 ± 0.04 of baseline with PV-IN versus 0.40 ± 0.06 of baseline with SST-IN photostimulation; p < 0.01 for both comparisons, using mixed-effects model analyses). To ensure that this difference did not simply reflect a greater number of activated PV-INs, we quantified the number of ChrimsonR-expressing PV-INs versus SST-INs: in fact, there were roughly twice as many labeled cells per field (40 μm slice, imaged at 10 x) in the $Sst^{Cre}$ mice (10.0 ± 0.7 SST-INs versus 5.5 ± 0.6 PV-INs; p < 0.001; unpaired t-test; n = 12, 17 fields). These data show that PV-INs are particularly effective at preventing PP-evoked GC activation.

## Discussion

Prior work has established that Dravet syndrome (DS) is largely due to heterozygous pathogenic loss-of-function variants in *SCN1A* leading to haploinsufficiency of the voltage-gated sodium channel α subunit Nav1.1 (*Claes et al., 2001*). However, mechanisms by which loss of Nav1.1 leads to the epilepsy, neurodevelopmental disability and SUDEP that clinically defines DS remains under intense investigation. Work in preclinical experimental mouse models of DS ($Scn1a^{+/-}$ mice) demonstrates temporal lobe-onset seizures (*Liautard et al., 2013*) and prominent activation of hippocampal DG during temperature-sensitive seizures (*Dutton et al., 2017*), suggesting involvement of hippocampus in seizure generation in $Scn1a^{+/-}$ mice. This corresponds with human clinical data demonstrating hippocampal involvement in patients with *SCN1A*-related seizure disorders (*Colosimo et al., 2007*; *Gaily et al., 2013*; *Van Poppel et al., 2012*). In this study, we employed 2 P calcium imaging in acute slice in an experimental mouse model of DS (*Figure 1*) to demonstrate a profound abnormality in a corticohippocampal circuit element strongly linked to temporal lobe seizures and epilepsy (*Figure 2*). This abnormality was not prominent in early postnatal mice (P14-21) yet became more profound in the chronic phase of the disorder. In contrast, defects in PV-IN excitability became less pronounced at this later timepoint (*Figure 3*) and hence could not account for the identified circuit dysfunction (*Figure 4*). Indeed, $Scn1a^{+/-}$ mice had no impairment in disynaptic feedforward inhibition from entorhinal cortex onto GCs; rather, there was increased strength of monosynaptic excitatory input from entorhinal cortex to DG, with evidence of increased release probability implicating pre-synaptic mechanisms of PP dysfunction driving circuit dysregulation (*Figure 5*). We then extended our study in vivo and found that both optogenetic activation of the perforant path (*Figure 6*) and chemogenetic inhibition of PV-INs (*Figure 7*) significantly decreased the temperature threshold for seizure generation in $Scn1a^{+/-}$ mice. Finally, we targeted and corrected this circuit abnormality in vitro by recruiting inhibitory reserve via optogenetics in slice (*Figure 8*). Taken together, these results demonstrate a clear locus of dysfunction despite complex circuit abnormalities and suggest unexpected plastic rearrangements

in corticohippocampal circuity that may relate to the dynamic clinical evolution and maintenance of chronic epilepsy and intellectual disability in DS.

## Calcium imaging enables large-scale comparison of dentate gyrus activation in wild-type and *Scn1a+/-* mice

2 P imaging allows simultaneous read-out of the activity of hundreds (and now even many thousands or more *Stringer et al., 2019*) of neurons, with cellular resolution and at high speeds. Our aim was to use 2 P imaging for a large-scale comparison of GC activation between genotypes (*Scn1a⁺/⁻* versus wild-type), which is a valid approach only if there is no genotype difference in the relationship between the number of action potentials that a neuron fires and its resulting calcium signal (d$F$/$F0$). This is indeed a theoretical concern as previous studies have demonstrated dysregulation of voltage gated calcium channel expression patterns in hippocampal neurons in rodent models as well as human patients with TLE (*Xu and Tang, 2018*; *Yaari et al., 2007*). Expression of calcium binding proteins, such as calmodulin and calbindin, may also be altered by, and play a role in, the pathological process of epilepsy (*Xu and Tang, 2018*). We therefore directly tested the ability of GCaMP7s to report action potentials in both *Scn1a⁺/⁻* and wild-type GCs (*Figure 1*), confirming that we could detect single action potentials in both genotypes under our experimental conditions and that there was no difference between genotypes in the relationship between action potentials and calcium-induced fluorescence increases. That this was the case may relate to the relative homogeneity of GCs and the fact that intrinsic GC cellular physiology does not appear to change in *Scn1a⁺/⁻* mice (*Figure 2—figure supplement 4*; *Table 1*).

## Corticohippocampal circuit hyperexcitability and ictogenesis in *Scn1a+/-* mice

Consistent with previous literature (*Yu et al., 2013*), we found that, in wild-type mice, the response of GCs to PP stimulation is robust in early postnatal mice (P14-21), becoming sparse by the young adult timepoint (*Figure 2*). This developmental pattern was disrupted in *Scn1a⁺/⁻* mice: the GC response was only mildly abnormal via selected measures in early postnatal mice (P14-21), whereas relative to wild-type age-matched controls, GCs were profoundly hyper-activated in young adult mice across all conditions and analyses.

A central mechanistic hypothesis in acquired temporal lobe epilepsy (TLE) is that epileptogenesis results from uncontrolled excitatory input from EC to vulnerable downstream hippocampal areas, leading to damage to CA3 and propagation of epileptiform activity throughout the hippocampal network (*Behr et al., 1998*; *Dengler et al., 2017*). Our findings additionally implicate this corticohippocampal circuit in ictogenesis in the *Scn1a⁺/⁻* mouse model during the chronic phase. Specifically, we found that either challenging the circuit (via entorhinal cortical stimulation; *Figure 6*) or further disrupting it (via DG PV-IN inhibition; *Figure 7*) significantly exacerbated hyperthermia-evoked seizures in *Scn1a⁺/⁻* mice. Thus, corticohippocampal circuit hyperexcitability may be a convergent circuit-level feature shared across acquired TLE and DS in the chronic phase, although the inciting insult is completely different between these epilepsy types. Firm establishment of such a shared mechanistic link may require further investigation.

In chronic acquired TLE after brain injury (such as due to status epilepticus induced by a chemoconvulsant agent) in rodents, the current operating model is a transient (weeks) impairment of corticohippocampal circuit function during the so-called latent period, followed by reconstitution of normal/near-normal operations, then secondary temporoammonic (TA) pathway failure and/or chronic impairment of DG activity (*Ang et al., 2006*; *Dengler et al., 2017*). We did not directly investigate the TA pathway in this study, although our in vivo optogenetic stimulation of cell bodies within entorhinal cortex (*Figure 6*) likely activated TA as well as PP projections to hippocampus. Disambiguating the relative contribution of PP and TA pathway to ictogenesis in *Scn1a⁺/⁻* mice would be an interesting future direction.

Note that it is highly unlikely that this corticohippocampal circuit hyperexcitability provides a complete mechanistic explanation for seizures in *Scn1a⁺/⁻* mice, which places several important limitations on the scope of our interpretation. First, we only tested the impact of this circuit on hyperthermia-evoked seizures, and it is possible that spontaneous seizures are mechanistically dissimilar. Second, spontaneous seizures in this model begin at approximately P16-18 (*Hawkins et al., 2017*), at which time this circuit is only mildly hyperexcitable (*Figure 2*); hence, this circuit abnormality likely does

not underlie epilepsy onset in these mice (which may instead relate to the transient but profound hypoexcitability of PV-INs at early developmental time points), although it may contribute to ongoing seizures in the chronic phase. Finally, although there is evidence of temporal lobe dysfunction (*Gaily et al., 2013*; *Kasperaviciute et al., 2013*; *Siegler et al., 2005*; *Tiefes et al., 2019*; *Van Poppel et al., 2012*) in human patients with DS, DS is a multifocal epilepsy and seizures do not exclusively emanate from the temporal lobe.

## Mechanisms of corticohippocampal hyperexcitability in DS: expanding beyond the interneuron hypothesis

As PV-INs in DG have been shown to powerfully regulate the GC response to entorhinal cortical input (*Ewell and Jones, 2010*; *Lee et al., 2016*) and GABAergic interneurons are known to be abnormal in *Scn1a*$^{+/-}$ mice (*Bechi et al., 2012*; *Mistry et al., 2014*; *Ogiwara et al., 2013*; *Tsai et al., 2015*; *Yu et al., 2006*), we initially hypothesized that PV-IN dysfunction would be the primary mechanism underlying the GC hyperactivation observed here (*Figure 2*). However, our findings were not compatible with this conclusion, for multiple reasons.

First, we did observe severe impairment of *Scn1a*$^{+/-}$ DG PV-INs at the early postnatal timepoint (*Figure 3*; *Table 2*), consistent with previous studies characterizing hippocampal PV-INs in younger (≤ P21) mice (*Hedrich et al., 2014*; *Tsai et al., 2015*). However, the PP stimulation-evoked disynaptic feedforward IPSCs onto GCs were not impaired at this timepoint (*Figure 5—figure supplement 2*), suggesting that the observed differences at the rightward extreme of the *I*-F curve may be less physiologically relevant to DG circuit function.

Second, our analysis of *Scn1a*$^{+/-}$ DG PV-INs at the young adult timepoint showed only a subtle impairment in action potential generation, characterized by run-down of action potential peak during prolonged trains at high frequency in response to large-amplitude current injections, likely due to accumulation of sodium channel inactivation (*Figure 3*; *Table 2*). This near-normalization of DG PV-INs in young adult *Scn1a*$^{+/-}$ mice (*Figure 3*; *Table 2*) is consistent with prior work demonstrating that PV-IN dysfunction (in layer 2/3 barrel cortex) is limited to early developmental time points (*Favero et al., 2018*). Our finding is also consistent with recent work demonstrating that sensorimotor cortex PV-INs in adult *Scn1a*$^{+/-}$ mice exhibit normal activity levels during quiet wakefulness in vivo (*Tran et al., 2020*).

Third, we observed durable hyperactivation of GCs in response to PP activation in *Scn1a*$^{+/-}$ mice relative to wild-type even in the presence of GABA$_A$ receptor blockade (using PTX; *Figure 5E*), which argues against impaired inhibition as the explanation for the genotype difference. In contrast, in a prior study showing increased evoked GC activation following status epilepticus-induced TLE, dentate hyperactivation in the epilepsy condition relative to control was eliminated by PTX (*Dengler et al., 2017*, p.201). This illustrates how disparate underlying epilepsy types (genetic mutation versus brain injury) could converge at the level of the circuit (DG hyperactivity), yet via disparate cellular and/or synaptic mechanisms. This compelling idea requires further investigation.

Finally, we found that, although application of the Nav1.1-specific activator Hm1a corrected PV-IN cellular dysfunction, it failed to normalize the PP-evoked circuit-level response across GCs (*Figure 4*). This highlights the conclusion that, by the young adult time point, the mechanism underlying abnormal circuit-level corticohippocampal dysfunction is more complex than Nav1.1 insufficiency (although Hm1a application does not act by increasing sodium current density *Osteen et al., 2017*). This distinction may have treatment implications: recent preclinical work using antisense oligonucleotide (ASOs) to increase *Scn1a* expression showed promising improvement in the disease phenotype when delivered at P2 or P14, prior to seizure onset (*Han et al., 2020*). However, it remains to be seen whether boosting Nav1.1 expression with ASOs or via another approach would be effective in the chronic phase of the disorder, after secondary circuit deficits may already be in place. Another recent study (*Yamagata et al., 2020*), using CRISPR/dCas9-based gene activation to increase Nav1.1 expression in GABAergic interneurons at P28, found an increased temperature threshold of seizure induction; however, the treatment cohort had a marked increase in early mortality (prior to intervention at P28) versus other groups, rendering comparisons of the surviving mice difficult to interpret.

If inadequate inhibition does not underly the *Scn1a*$^{+/-}$ corticohippocampal circuit hyperexcitability, what does? Our measurement of evoked post-synaptic currents onto GCs unexpectedly revealed increased synaptic excitatory PP input to GCs in *Scn1a*$^{+/-}$ mice (*Figure 5A–C*). We found no genotype difference in GC response to puffed glutamate (*Figure 5F–G*; *Figure 5—figure supplement 1*). This

argues against a post-synaptic locus of the observed genotype difference, although the measured response to exogenous application of glutamate could be mediated by synaptic and/or extrasynaptic receptors. Future experiments, such as measurement of miniature EPSC amplitude and frequency, or use of strontium to probe presynaptic release, could further bolster this conclusion. Notably, we observed a lower PPR with repetitive stimulation (*Figure 5H–I*; *Figure 5—figure supplement 1*). This may suggest a higher pre-synaptic release probability at excitatory PP synapses in young adult $Scn1a^{+/-}$ mice relative to age-matched wild-type control. However, we cannot exclude a role for differences in relative recruitment of lateral versus medial PP fibers, which are known to exhibit different short-term synaptic dynamics (*Petersen et al., 2013*).

Taken together, our findings demonstrate excessive excitation in the cortico-hippocampal circuit that may reflect an increase in pre-synaptic release probability. The genotype differences in EPSC/IPSC ratio, evoked EPSC amplitude, and PPR, were not seen at the early postnatal timepoint (*Figure 5—figure supplement 2*), providing correlative evidence that such changes are important for the emergence of severe circuit hyperexcitability seen in the chronic phase of the disorder (*Figure 2*). Our findings are consistent with previous studies showing hyperexcitability and increased synaptic excitation in downstream hippocampal CA1 (*Gu et al., 2014*; *Han et al., 2012*; *Hedrich et al., 2014*; *Liautard et al., 2013*), as well as one prior study (*Tsai et al., 2015*) in a related mouse model which showed increased release probability with PP stimulation as well as increased frequency of spontaneous EPSCs in GCs.

The mechanism underlying the developmental pattern of the corticohippocampal circuit deficit remains unclear. The typical development of the corticohippocampal circuit – in which PP-evoked DG firing becomes more sparse – is at least in part mediated by increases in local circuit inhibition (*Dieni et al., 2013*; *Yu et al., 2013*); thus, it is not simply the case that the $Scn1a^{+/-}$ circuit remains in an 'immature' state. Rather, persistent GC hyperactivation could reflect accumulated or repetitive insult from ongoing seizures, a complex interaction between genotype and seizure burden (*Dutton et al., 2017*; *Salgueiro-Pereira et al., 2019*), and/or a pathologic compensatory increase in synaptic strength of the PP in response to loss of Nav1.1. These possibilities could be further explored using experimental manipulations to exacerbate (*Dutton et al., 2017*; *Hawkins et al., 2017*; *Salgueiro-Pereira et al., 2019*) or ameliorate (*Hawkins et al., 2017*) the epilepsy phenotype of $Scn1a^{+/-}$ mice; subsequent testing of the excitability of the corticohippocampal circuit could help parse the contribution of the genotype (common across cohorts) versus, for example, the effect of ongoing seizures, to disease severity. Conversely, it would be of interest to focally rescue $Scn1a$ levels exclusively in dentate gyrus and/or entorhinal cortex of $Scn1a^{+/-}$ mice (*Colasante et al., 2020*), to test whether the corticohippocampal circuit would remain abnormal (assuming continued seizures involving the hippocampus). Finally, while we did not directly assess dendritic integration by GCs – which are known to exhibit strong voltage attenuation that modulates dendritic integration and contributes to their sparse activation in response to PP input (*Krueppel et al., 2011*) – we do show that (a) the intrinsic excitability of GCs (as measured at the soma) is unchanged in $Scn1a^{+/-}$ mice at both developmental time points and (b) the response to exogenous glutamate application is the same between genotypes (with attendant caveats noted above).

## Rescue of corticohippocampal circuit excitability via targeted recruitment of PV-INs

To test the impact of local inhibitory interneurons on the recruitment of GCs by PP input, we combined 2 P imaging with optogenetic stimulation with the red-shifted channelrhodopsin variant ChrimsonR (*Klapoetke et al., 2014*; *Figure 8*). This technique enabled us to perform simultaneous imaging (of anatomically defined GCs) and photostimulation (of genetically defined interneuron subtypes; that is PV-INs or SST-INs) within the same slice. We found that targeted PV-IN activation in acute slice significantly decreased GC activation in response to PP input (*Figure 8*).

Optogenetic activation of PV-expressing cells (and fibers) within hippocampus has been shown to truncate seizures in vivo in a mouse model of TLE (*Krook-Magnuson et al., 2013*). It remains to be seen whether a similar strategy may be successfully employed against temporal lobe-onset (or other) seizures in $Scn1a^{+/-}$ mice. Our data shows that, while DG PV-INs do exhibit minor abnormalities in spike generation during the chronic phase of the disorder, these cells remain capable of discharging short trains of action potentials at high frequency ( > 200 Hz for trains of 10+ action potentials) such

that there is sufficient inhibitory 'reserve' that would enable these cells to be exogenously recruited to balance aberrant excitatory activity.

Improved understanding of the mechanisms of ictogenesis in DS is critical toward the development of novel therapies for what is currently an incurable and treatment-resistant disorder associated with high mortality. Our results indicate complex network dysfunction attributed to excessive excitation – rather than exclusively due to dysfunction of GABAergic subsystems – in the chronic phase of the disease. We provide evidence that the corticohippocampal circuit is hyperexcitable in *Scn1a*$^{+/-}$ mice, as has been shown in animal models of acquired temporal lobe epilepsy (*Dengler et al., 2017*; *Krook-Magnuson et al., 2015*), suggesting that DG circuits play a key role in temporal lobe vulnerability to seizure onset and/or propagation. Thus, corticohippocampal circuits may be an attractive target for seizure suppression or interruption across diverse forms of epilepsy. Specifically, DG PV-INs powerfully regulate seizures and circuit excitability in *Scn1a*$^{+/-}$ mice and may thus be an attractive candidate population for targeted therapeutic intervention. As sparse GC firing is also thought to be crucial for cognitive functions subserved by the hippocampus, such as pattern separation and memory encoding (*Chung et al., 2021*; *Leutgeb et al., 2007*; *McHugh et al., 2007*; *Rolls, 2010*; *Treves et al., 2008*), correction of this circuit dysfunction may also potentially modulate the severe and chronic intellectual disability characteristic of DS and even more difficult to approach than epilepsy.

# Materials and methods

## Key resources table

| Reagent type (species) or resource | Designation | Source or reference | Identifiers | Additional information |
|---|---|---|---|---|
| Genetic reagent (*Mus musculis*) | 129S-Scn1atm1Kea/Mmjax | Jax | RRID:MMRRC_037107-JAX | Dr. Jennifer A. Kearney, Northwestern University |
| Genetic reagent (*M. musculis*) | B6;129P2-*Pvalb*$^{tm1(cre)Arbr/J}$ | Jax | RRID:IMSR_JAX: 017320 | |
| Genetic reagent (*M. musculis*) | B6J.Cg-*Sst*$^{tm2.1(cre)Zjh/MwarJ}$ | Jax | RRID:IMSR_JAX:028864 | |
| Genetic reagent (*M. musculis*) | Rosa-CAG-LSL-tdTomato | Jax | RRID:IMSR_JAX:007914 | |
| Genetic reagent (*M. musculis*) | C57BL/6 J | Jax | RRID:IMSR_JAX:000664 | |
| Recombinant DNA reagent | pGP-AAV-syn-jGCaMP7s-WPRE | Addgene | 104487-AAV9 | $3 \times 10^{13}$ cfu/mL |
| Recombinant DNA reagent | pAAV-Syn-ChrimsonR-tdT | Addgene | 59171-AAV9 | $2.3 \times 10^{12}$ cfu/mL |
| Recombinant DNA reagent | pAAV-CaMKIIa-hChR2(H134R)-EYFP | Addgene | 26969-AAV9 | $3.9 \times 10^{12}$ cfu/mL |
| Recombinant DNA reagent | AAV9-CamKIIα-eYFP-WPRE-hGH | UNC Vector Core | | $3.9 \times 10^{12}$ cfu/mL |
| Recombinant DNA reagent | pAAV-hSyn-DIO-hM4D(Gi)-mCherry | Addgene | 44362-AAV9 | $9.0 \times 10^{12}$ cfu/mL |
| Peptide, recombinant protein | Hm1a | Alomone | STH-601 | 50 nM |
| Chemical compound, drug | Quantum dot | PMID:25326662 | | Igor L. Medintz, U.S. Naval Research Laboratory |
| Chemical compound, drug | Picrotoxin | Tocris Bioscience | 11–281 G | 100 µM |
| Chemical compound, drug | Clozapine N-oxide (CNO) | Sigma-Aldrich | C0832 | 10 µM (slice); 10 mg/kg (in vivo) |
| Software, algorithm | pClamp 10 | Clampfit | RRID:SCR_011323 | V10.0 |
| Software, algorithm | Matlab | Mathworks | RRID:SCR_001622 | 2019 a |
| Software, algorithm | Python Programming Language | http://www.python.org/ | RRID:SCR_008394 | |
| Software, algorithm | Analysis of two photon imaging data | This paper *Somarowthu, 2022* | https://github.com/GoldbergNeuroLab/Mattis-et-al.-2022 | |
| Software, algorithm | Analysis of whole cell patching data | This paper *Evans, 2022*; *Goff, 2022* | https://github.com/GoldbergNeuroLab/Mattis-et-al.-2022 | |

*Continued on next page*

*Continued*

| Reagent type (species) or resource | Designation | Source or reference | Identifiers | Additional information |
|---|---|---|---|---|
| Antibody | anti-Parvalbumin antibody (Rabbit polyclonal) | Swant Cat# PV27 | RRID:AB_2631173 | (1:1000) |
| Antibody | anti- Somatostatin antibody (Rat monoclonal) | Millipore Cat# MAB354 | RRID:AB_2255365 | (1:50) |
| Antibody | Anti-Rat IgG (H + L) Antibody, Alexa Fluor 488 (Goat polyclonal) | Molecular Probes Cat# A-11006 | RRID:AB_141373 | (1:500) |
| Antibody | Anti-Rabbit IgG (H + L) Antibody, Alexa Fluor 488 (Goat polyclonal) | Molecular Probes Cat# A-11008 | RRID:AB_143165 | (1:500) |
| Other | DAPI stain | Thermo Fisher Scientific Cat# D1306 | RRID:AB_2629482 | (1:50,000) |

All experiments were carried out as per protocols approved by the Institutional Animal Care and Use Committee at the Children's Hospital of Philadelphia, in accordance with ethical guidelines set forth by the National Institutes of Health.

## Experimental animals

Male and female mice were used in equal proportion. Mice were weaned at P21 and were subsequently group-housed to the extent possible in cages containing up to five mice. Mice were maintained on a 12 hr light/dark cycle with access to food and water ad libitum. All mice were genotyped via PCR analysis of a tail snip obtained at P7.

The $Scn1a^{+/-}$ mice used in this study have a targeted deletion of the first exon of the *Scn1a* gene, with a resulting null allele and 50% decrease in Nav1.1 protein (*Miller et al., 2014*, p. 20; *Mistry et al., 2014*). To generate mice for experimental use, male $Scn1a^{+/-}$ mice on a 129S6.SvEvTac background (RRID: MMRRC_037107-JAX) were crossed to female mice on a C57BL/6 J background, either wild-type C57BL/6 J mice (RRID:IMSR_JAX:000664) or Cre-driver lines (see below). The resulting progeny have a mixed 50:50 129S6:BL6/J background, on which the DS phenotype has been extensively characterized (*Han et al., 2020*; *Hawkins et al., 2017*; *Miller et al., 2014*; *Mistry et al., 2014*).

For experiments requiring targeted viral expression in, or visualization of, PV-INs, female $Pvalb^{Cre}$.tdT double-heterozygous mice were generated from a cross between homozygous tdTomato reporter/Ai14D mice (Rosa-CAG-LSL-tdTomato; RRID:IMSR_JAX:007914) and homozygous $Pvalb^{Cre}$ mice (B6;129P2-$Pvalb^{tm1(cre)Arbr}$; RRID:IMSR_JAX:008069). These mice are on a 100% C57BL/6 J background. The F1 progeny include $Scn1a^{+/-}.Pvalb^{Cre}$.tdT and $Pvalb^{Cre}$.tdT mice (1:8 predicted Mendelian ratio for each). These mice express tdT in PV-INs, which facilitates targeted experiments. A similar strategy was used for experiments targeting viral expression to SST-INs, with $Sst^{Cre}$ mice (B6J.Cg-$Sst^{tm2.1(cre)Zjh}$; RRID:IMSR_JAX:028864).

## Viral injections

Subdural viral injections (early postnatal mice): To achieve GCaMP expression in early postnatal mice, subdural viral injections (*Gao et al., 2019*) were performed on mice at age P0-2. Mice were anesthetized by cooling on ice until cessation of motor activity. The scalp was sterilized using ethanol. Virus was injected halfway between lambda and bregma and halfway between the midline and the eye, using a 10 µL syringe (Hamilton) and a 33 G beveled needle (World Precision Instruments). Injections of GCaMP (AAV9.syn.GCaMP7s.WPRE; Addgene) were performed bilaterally using 1 µL of virus at a genomic titer of $1.5 \times 10^{13}$ cfu/mL.

Stereotaxic viral injections (young adult mice): Mice were anesthetized by inhalation of isoflurane (4% induction, 1–2% maintenance) in oxygen. Anesthesia depth was monitored by response to toe pinch and by breathing rate. Animals were injected subcutaneously with buprenorphine (0.5–1.0 mg/kg, sub-Q), cefazolin (500 mg/kg, sub-Q), and meloxicam (5 mg/kg, sub-Q), for post-operative analgesia and anti-bacterial prophylaxis. Mice were placed in a stereotaxic apparatus (Kopf Instruments) while resting on a heating pad. After removal of fur and sterilization of the scalp, a scalpel was used to expose the skull, and target regions were identified in relation to skull surface landmarks (lambda and bregma). Craniotomies were made using a hand-held drill. Injections were performed with 200–250 nL of virus per site, at a rate of 100 nL/min, using a 10 µL syringe (Hamilton) and a 33 G

beveled needle (World Precision Instruments). Injection coordinates for entorhinal cortex (relative to Bregma, in mm) were –4.5 (A/P) / 3.0 (M/L) / 4.5 (D/V); –4.5 (A/P) / 3.6 (M/L) / 4.2 (D/V); and –4.8 (A/P) / 3.2 (M/L), 3.2 and 3.8 (D/V). Dorsal and ventral DG were targeted via 10–12 injections per animal, spanning from –1.5 to –3.8 A/P, 0.7–2.6 M/L, and 2.0–3.8 D/V. Injections were unilateral (R hemisphere) except for mice used for in vivo chemogenetics experiment, which had bilateral injections. The viruses and final genomic titers (following dilution) were as follows: AAV9-syn-GCaMP7s-WPRE (Addgene 104487; $3 \times 10^{12}$ cfu/mL); AAV9-syn-FLEX-ChrimsonR-tdT (Addgene 59171; $2.3 \times 10^{12}$ cfu/mL); AAV9-CamKIIα-hChR2(H134R)-eYFP-WPRE-hGH (Addgene 26969; $3.9 \times 10^{12}$ cfu/mL); AAV9-CamKIIα-eYFP-WPRE-hGH (UNC Vector Core; $3.9 \times 10^{12}$ cfu/mL); AAV8-hSyn-DIO-hM4D(Gi)-mCherry (Addgene 44362; $9.0 \times 10^{12}$ cfu/mL). All experiments were performed at least 3 weeks after injection to allow for virus expression.

## Acute slice preparation

Mice were deeply anesthetized using inhaled isoflurane. Adult mice were perfused transcardially with 5 mL ice-cold sucrose-based cutting solution. Brains were removed and immediately transferred to ice-cold sucrose solution (in mM: NaCl, 87; sucrose, 75; KCl, 2.5; CaCl$_2$, 1.0; MgSO$_4$, 2.0; NaHCO$_3$, 26; NaH$_2$PO$_4$, 1.25; glucose, 10), and equilibrated with 95% O$_2$ and 5% CO$_2$. Hippocampal-entorhinal cortex (HEC) slices were obtained to preserve within the slice the PP connection between the entorhinal cortex and the hippocampus (*Xiong et al., 2017*). A mid-sagittal cut was made to divide the hemispheres, and one hemisphere was mounted onto an angled agar block. The blade therefore passed through the block at an acute (15°) downward angle from rostral to caudal, resulting in modified horizontal/axial HEC slices. Slices were sectioned at a 300 µm thickness using a Leica VT-1200S vibratome. Slices were transferred to a holding chamber containing the same sucrose solution. After a recovery period of 30 min at 30°C–32°C, the holding chamber was removed from the water bath and allowed to return to room temperature.

## Perforant path stimulation

The perforant path was stimulated using a 125 µm diameter concentric bipolar microelectrode (FHC Inc NC0950490). The electrode was placed in the entorhinal cortex adjacent to the apex of the dentate gyrus granule cell layer and approximately 100 µm from the hippocampal fissure so as not to directly stimulate GCs. Since GCs are known to be more readily activated by PP stimulation in younger mice (*Yu et al., 2013*), we selected different ranges of stimulation intensity for the two groups (25–400 µA for early postnatal; 100–500 µA for young adult).

## Slice electrophysiology

Slices were transferred to a recording chamber (shielded from light in the case of opsin-containing slices) and continuously perfused with oxygenated artificial cerebrospinal fluid (ACSF; in mM: NaCl, 125; KCl, 2.5; CaCl$_2$, 2.0; MgSO$_4$, 1.0; NaHCO$_3$, 26; NaH$_2$PO$_4$, 1.25; glucose, 10), continuously bubbled with 95% O$_2$ and 5% CO$_2$, at a rate of approximately 3 mL/min. All slice physiology experiments were performed at 30°C–32°C.

DG GCs were identified by localization within the GCL, morphology, and electrophysiological discharge pattern. PV-INs were identified by endogenous tdT expression visualized with epifluorescence and were typically large cells located at the border between the granule cell layer and the hilus.

Whole cell voltage- and current-clamp recordings were obtained using borosilicate glass electrodes, pulled using a P-97 puller (Sutter Instruments) for a tip resistance of 3–4 MΩ for voltage clamp and 4–6 MΩ for current clamp. For characterization of intrinsic electrophysiological properties, the pipette solution was a K-Gluconate solution that contained, in mM: K-gluconate, 130; KCl, 6.3; EGTA, 0.5; MgCl$_2$, 1.0; HEPES, 10; Mg-ATP, 4.0; Na-GTP, 0.3; pH adjusted to 7.30 with KOH and osmolarity adjusted to 285 mOsm with 30% sucrose. For measurement of EPSCs and IPSCs in GCs, whole cell recordings were obtained in an independent set of GCs with a cesium-based pipette solution that contained, in mM: Cs-methylsulfonate, 125; HEPES, 15; EGTA 0.5; Mg-ATP, 2.0; Na-GTP, 0.3; phosphocreatine-Tris2, 10; QX 314, 2.0; TEA, 2; pH adjusted to 7.35 with CsOH and osmolarity adjusted to 295 mOsm with 30% sucrose.

In a subset of recordings, exogenous L-glutamic acid (Sigma) was diluted in ACSF (1 mM) and applied using a brief pressure pulse (5, 10, or 30ms duration; 6 PSI) delivered by a Picospritzer III

(Parker Instrumentation) connected to a patch pipette (resistance ~2–4 mΩ) positioned within the mid-granule cell layer. A minimum of 3 locations within the molecular layer were trialed for each recording, and the maximal responses were recorded. Responses were measured in GCs held at –70 mV in voltage-clamp.

For cell-attached recordings, Quantum Dots (*Andrásfalvy et al., 2014*) were employed to enable visualization of the pipette tip via two-photon imaging. Dried Quantum Dots (625 nm emission) were reconstituted in hexane. Freshly pulled borosilicate glass electrodes (7–9 MΩ) were dipped 5–10 times into the solution, while maintaining positive pressure to avoid clogging the pipette tip. A subset of cell-attached recordings was performed with 100 µM picrotoxin added to the bath solution to increase the proportion of cells responsive to PP stimulation.

Signals were sampled at 50 kHz, amplified with a MultiClamp 700B amplifier (Molecular Devices), filtered at 10 kHz, digitized using a DigiData 1550, and acquired using pClamp10 software. Data were not included for final analysis if the cell had a membrane potential greater (less negative) than –60 mV (for GCs) or greater than –50 mV (for PV-INs), or if access resistance was greater than 30 MΩ. Series resistance compensation (bridge balance) was applied throughout current clamp experiments with readjustments as necessary. Reported values for membrane potential and AP threshold are not corrected for the liquid junction potential.

## 2P calcium imaging and combined optogenetic stimulation in acute slice

Slices were continuously perfused with oxygenated ACSF (as for slice physiology, above) and were heated to 30–32°C. Imaging was performed using a customized two-photon laser scanning microscope (Bruker Ultima) equipped with a resonant scanner (Cambridge Technologies) and a MaiTai DeepSee Ti:Sapphire mode-locked pulsed infrared laser (SpectraPhysics). GCaMP7s was imaged at 920 nm with a gallium arsenide phosphide (GaAsP) photodetector (H7422-40; Hamamatsu) through a 25 X/0.95-NA water immersion objective (Leica). Laser power for imaging in the acute brain slice preparation typically ranged from 10 to 20 mW (at the specimen). For each stimulation, 500 frames were collected with a frame period of 33.8ms (30 Hz) and a resolution of 512 × 512 pixels.

Full field photostimulation was delivered through the objective lens with a high-powered red (660 nm) LED (M660L4; Thorlabs). This was mounted to the epifluorescence port of the microscope and routed to the sample below the PMTs using a custom notched dichromic mirror/polychromic beam splitter with reflectance from 660 ± 20 nm (ZT660/40; Chroma). LEDs were controlled via an LED driver (LEDD1B; Thorlabs) driven by a TTL pulse from and synchronized with either the imaging acquisition software (PrairieView) or electrophysiology data acquisition software (pClamp). Irradiance at the specimen was measured through the imaging objective using a photodiode power sensor (Thorlabs, S120C) and was 26 mW/mm$^2$ for all experiments.

## Slice pharmacology

Picrotoxin (Tocris) was used at a saturating concentration of 100 µM in ACSF. Hm1a (Alomone Labs STH-601) was used at a 50 nM final concentration in 0.025% BSA in ACSF. CNO (Sigma) was reconstituted in 4% DMSO and was used at 10 µM in ACSF. Drugs were perfused at 3 mL/min after a baseline recording was obtained.

## Immunohistochemistry

Mice were deeply anesthetized with isoflurane and transcardially perfused with ice-cold PBS followed by 4% paraformaldehyde (PFA) in PBS. Brains were removed, post-fixed in PFA overnight, and equilibrated in a 30% sucrose solution. 40 µm sections were obtained using a frozen sliding microtome (American Optical). Slices were placed in cryoprotectant (25% glycerol, 30% ethylene glycol, 45% PBS, pH adjusted to 6.7 with HCl) for long-term storage. Before immunostaining, slices were washed in PBS to remove the cryoprotectant and then blocked and permeabilized for one hour at room temperature with 0.3% Triton X-100 (Sigma) and 3% normal goat serum (NGS) in PBS. We incubated slices for ~48 hr at 4° C with a primary antibody for either parvalbumin (Swant PV27, 1:1000) or somatostatin (Millipore MAB354, 1:50) in PBS with 0.3% Triton X-100% and 3% NGS. Slices were then washed with PBS and incubated with a secondary antibody in PBS with 0.3% Triton X-100% and 1% NGS: Alexa Fluor 488 goat anti-rat for PV staining (Molecular Probes, A11006) or goat anti-rabbit for SST staining

(Molecular Probes, A11034). Slices were washed, labeled with DAPI (1:50,000; Fisher Scientific), and cover-slipped using polyvinyl alcohol mounting medium with DABCO (Sigma 10981). Imaging was performed using an upright epi-fluorescence microscope (Nikon Instruments).

## Hyperthermic seizure generation

Seizures were elicited by induction of hyperthermia. Mouse body temperature was monitored continuously via a rectal probe (Physitemp), which was secured in place such that stable recordings were obtained while mice moved freely throughout the recording chamber. Mice were placed in a room temperature chamber and were monitored for 5 min to establish a stable body temperature. Mice were then transferred to a second chamber in which they were passively heated via gradual temperature ramp using an overhead lamp. Heating was continued either until a behavioral seizure was observed (at which point the exact body temperature was noted and recorded), or until the body temperature reached 42.5 °C, upon which the mouse was immediately removed from the chamber and cooled using ice.

## In vivo optogenetics

Following stereotaxic virus injection, mice were implanted with a fiberoptic cannula (200 μm core diameter, 240 μm outer diameter, 0.22 NA, flat tip) coupled to a Zirconia ferrule (1.25 mm outer diameter; Doric Lenses). This was implanted above the entorhinal cortex (unilateral, on the right hemisphere). Implant coordinates (relative to Bregma) were –4.5 (A/P); 3.0 (M/L); 3.9 (D/V). The implant was secured to the skull with a layer of adhesive cement (C&B metabond) followed by dental cement (Patterson dental) and then capped prior to use.

In vivo optogenetic stimulation was performed approximately 3 weeks after surgery to allow for virus expression and recovery from surgery. Blue light was generated by a 473 nm laser (Shanghai Laser & Optics Century Co.) and delivered through a fiberoptic patch cord (0.22 NA, 200 μm core diameter; Doric Lenses) coupled to the implanted fiberoptic cannula via a connecting plastic sleeve (Precision Fiber Products). Blue laser output was controlled using a pulse generated (Master-8) to deliver 5 ms light pulse trains at a rate of 20 Hz, 5 s on / 5 s off. Light power was 3 mW, measured at the tip of the fiberoptic.

## In vivo chemogenetics

CNO was reconstituted in 4% DMSO (in saline) and stored in a 10 mM stock solution. The stock solution was thawed immediately prior to use, diluted 1:2 (in saline), and injected subcutaneously at 10 mg/kg. The vehicle control was 2% DMSO in saline. Mice were injected intraperitoneally 30 min prior to seizure induction. Consecutive seizures were triggered 24 hr apart.

## Analysis of whole-cell electrophysiology data

All analysis (*Evans, 2022*; *Goff, 2022*) was performed blind to genotype using Matlab (Mathworks) or Python (custom software using Python 3.7 and the pyABF model) with quality control using manual confirmation in Clampfit (pCLAMP). Resting membrane potential ($V$m) was measured directly using the average value of a 1 s sweep with no direct current injection. Input resistance ($R$m) was calculated using the average response to a small hyperpolarizing step near rest, using $R$m = $\Delta V/I$ for each sweep. Membrane time constant was calculated from the single exponential fit to the hyperpolarizing response to a negative current injection. Rheobase was determined as the minimum current injection that elicited APs using a 600ms sweep at 10–25 pA intervals. AP threshold was calculated as the value at which the derivative of the voltage (d$V$/dt) first reached 10 mV/ms. AP peak refers to the absolute maximum voltage value of an individual AP. AP amplitude was the difference between AP peak and AP threshold; AP rise time was the time from AP threshold to AP peak. AP half-width is defined as the width of the AP (in ms) at half-maximal amplitude (half the voltage difference between the AP threshold and peak). AHP amplitude was calculated as the depth of the after-discharge potential (in mV) relative to AP threshold. Sag (produced by $I$h), was calculated as the steady state relative to maximal hyperpolarization in response to a negative current injection, expressed as a percentage. Maximal instantaneous firing was calculated as the inverse of the smallest inter-spike interval elicited during a suprathreshold 600ms current injection. Maximal steady-state firing was defined as the maximal mean firing frequency during a suprathreshold 600ms current injection. Spikes were defined

as having a clear threshold, at which the derivative of the voltage (d$V$/dt) is greater than 10 mV/ms, a spike height overshooting 0 mV, and an amplitude of at least 40 mV. All quantification of single spike properties was done at the first spike to the current injection twice rheobase. $I$-f plots were created using the steady-state firing calculated for each current step.

Paired-pulse ratio was defined as the ratio of the amplitude of the second EPSC to amplitude of the first EPSC (extrapolating a new baseline following the prior EPSC), in response to 20 Hz perforant path stimulation.

## Analysis of 2P imaging data

Analysis was performed using custom code (*Somarowthu, 2022*) in Matlab. All conditions under each field of view were grouped together as a raw data file. Motion was minimal or absent in most cases, but there was some drift during longer imaging sessions, such as with pharmacologic application. Therefore, within each raw data file, all individual frames were motion corrected using the Non-Rigid Motion Correction (NoRMCorre) Matlab toolbox (*Pnevmatikakis, 2019*). The first 200 frames were used for the template. The motion corrected individual frames were averaged to obtain an average image for the respective field of view. Regions of interest (ROIs) were identified from the average image. ROIs were individual GCs, and were identified using NeuroSeg (*Guan et al., 2018*), a Matlab toolbox with both automatic and manual options for selection of regions of cells; we primarily relied upon the manual option as we observed that the automated algorithm was not able to effectively identify the densely packed GCs.

The raw traces from each identified cell were extracted and low-pass filtered. The first 50 frames of each trace in each condition were considered as a baseline to compute d$F$/$F0$ for each trace. Peaks of the traces were detected using the findpeaks function in Matlab. A threshold of (mean + 3*standard deviation) was used to define significant peaks. A cell was detected as active if a significant peak is shown in the respective trace for that particular condition. The results were summarized for each condition by both the proportion of active cells and the individual significant peak amplitudes.

A mixed modeling approach was implemented for statistical analysis of data across populations. This was performed due to the hierarchical experimental design, in which sources of variance could arise from effects of imaging fields, slices, and/or mice. The proportion of activated cells were modeled as a binomial distribution with a probit link function. The amplitude of peaks was modeled as a normal distribution with a logarithmic link function.

For analyses involving comparisons within a given imaging field (e.g. response to PP stimulation alone versus PP stimulation plus optogenetic activation of PV-INs), mixed modeling was used to test the hypothesis that any difference between the paired data is significantly different from zero; if the difference between pairs is significantly different than zero, that implies the paired data are significantly different from each other. Mouse, slice, field, and cell information are again used as the random factors in the mixed modeling.

Some analyses ('responds to each stim') include for each condition only the cells that are active that condition. Other analyses ('responds to max stim') include for every condition all cells that are active under maximal stimulation conditions.

For spike deconvolution, spikes were extracted using constrained non-negative matrix factorization (*Pnevmatikakis et al., 2016*). The deconvolution parameters such as rise time and decay time were fine-tuned based on the ground truth data. A cell was defined as active if there were one or more spikes detected. Deconvolution yielded a single spike or multiple spikes based on the perforant path stimulation pattern (1 pulse versus four pulses). Irrespective of the stimulation, the total deconvolved spikes are derived by summing the individual deconvolved spikes obtained during the total time range of stimulation.

## Acknowledgements

This work was supported by NIH NINDS Research Education Grant (R25) and NIH NINDS K08 NS121464 to JM, a Postdoctoral Fellowship Award from the Dravet Syndrome Foundation to AS, as well as NIH NINDS K08 NS097633, NIH NINDS R01 NS110869, the Dana Foundation David Mahoney Neuroimaging Program research grant, and the Burroughs Wellcome Fund Career Award for Medical Scientists to EMG. This work was supported in part by the Institute for Translational Medicine and Therapeutics of the University of Pennsylvania via the National Center for Advancing

Translational Sciences under NIH UL1TR001878. We would also like to thank the Women's Committee of The Children's Hospital of Philadelphia for prior support. Statistical support was obtained from the Center for Human Phenomic Science (CHPS; CTSA grant UL1TR001878). We thank Colin Evans for contributing Python code for analysis of electrophysiology data. We thank the GENIE project and the Janelia Research Campus of the HHMI for distribution of GCaMP7, Ed Boyden (MIT) for distribution of ChrimsonR, and Igor L Medintz (U.S. Naval Research Laboratory) for the gift of quantum dots.

## Additional information

### Funding

| Funder | Grant reference number | Author |
|---|---|---|
| National Institute of Neurological Disorders and Stroke | R25 NS065745 | Joanna Mattis |
| National Institute of Neurological Disorders and Stroke | K08 NS097633 | Ethan M Goldberg |
| National Institute of Neurological Disorders and Stroke | R01 NS110869 | Ethan M Goldberg |
| Dana Foundation | David Mahoney Neuroimaging Grant | Ethan M Goldberg |
| Burroughs Wellcome Fund | Career Award for Medical Scientists | Ethan M Goldberg |
| National Institute of Neurological Disorders and Stroke | K08 NS121464 | Joanna Mattis |
| Institute for Translational Medicine and Therapeutics | Translational Biomedical Imaging Center (TBIC) | Joanna H Mattis Ethan M Goldberg |

The funders had no role in study design, data collection and interpretation, or the decision to submit the work for publication.

### Author contributions

Joanna Mattis, Conceptualization, Data curation, Formal analysis, Funding acquisition, Investigation, Methodology, Project administration, Writing – original draft, Writing – review and editing; Ala Somarowthu, Data curation, Formal analysis, Funding acquisition, Investigation, Software, Writing – review and editing; Kevin M Goff, Data curation, Formal analysis, Investigation, Software, Writing – review and editing; Evan Jiang, Jina Yom, Laura M Mcgarry, Huijie Feng, Keisuke Kaneko, Investigation; Nathaniel Sotuyo, Investigation, Methodology; Ethan M Goldberg, Conceptualization, Data curation, Formal analysis, Funding acquisition, Investigation, Methodology, Project administration, Resources, Supervision, Writing – original draft, Writing – review and editing

### Author ORCIDs

Joanna Mattis ⦿ http://orcid.org/0000-0003-0341-1270
Kevin M Goff ⦿ http://orcid.org/0000-0001-5862-0219
Keisuke Kaneko ⦿ http://orcid.org/0000-0002-5071-0057
Ethan M Goldberg ⦿ http://orcid.org/0000-0002-7404-735X

### Ethics

This study was performed in strict accordance with the recommendations in the Guide for the Care and Use of Laboratory Animals of the National Institutes of Health. All of the animals were handled according to approved institutional animal care and use committee (IACUC) protocol 21-001152 of The Children's Hospital of Philadelphia. All surgery was performed under isoflurane anesthesia, and every effort was made to minimize suffering.

Decision letter and Author response
Decision letter https://doi.org/10.7554/eLife.69293.sa1
Author response https://doi.org/10.7554/eLife.69293.sa2

## Additional files

### Supplementary files
• Transparent reporting form

### Data availability

All data generated or analyzed during this study are included in the manuscript and supporting files. Source data files for all Figures (1-8) and Tables (1-2) have been included and are also available via G-Node at: https://gin.g-node.org/GoldbergNeuroLab/Mattis-et-al-2022. Source code has been made available here: https://github.com/GoldbergNeuroLab/Mattis-et-al.-2022 (copy archived at swh:1:rev:a907129d2e81837a940c82c84af3a07c3200f510).

The following dataset was generated:

| Author(s) | Year | Dataset title | Dataset URL | Database and Identifier |
|---|---|---|---|---|
| Mattis JH, Somarowthu A, Goldberg EM | 2022 | Accompanying dataset for Mattis et al., 2022 | https://doi.org/10.12751/g-node.kxr89n | G-Node, 10.12751/g-node.kxr89n |

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
