## [Editor Report]

Recent work has shown that one of the major dogmas in epilepsy – that interneuron deficits underlie Dravet syndrome and maybe other epileptic encephalopathies – is overly simplistic. This manuscript makes a significant step forward with novel findings using ex vivo and in vivo experiments providing strong evidence that changes in excitatory connections in the corticohippocampal circuit contribute to mechanisms that drive epilepsy in Dravet syndrome.

---

## [Decision Letter]

**Decision letter after peer review:**

Thank you for submitting your article "Corticohippocampal circuit dysfunction in a mouse model of Dravet syndrome" for consideration by *eLife*. Your article has been reviewed by 3 peer reviewers, one of whom is a member of our Board of Reviewing Editors, and the evaluation has been overseen by Gary Westbrook as the Senior Editor. The reviewers have opted to remain anonymous.

Summary

The authors describe circuit changes in a hemizygous mouse model that potentially contribute to the severe epilepsy disorder Dravet syndrome. They show that hyperactivation of the dentate gyrus results from enhanced synaptic excitation rather than reduced synaptic inhibition, the latter being the widespread view in the field. They provide some evidence that this alteration occurs in the age range when inhibitory interneuron dysfunction has been normalized, providing an interesting potential mechanism by which neural circuit function is altered even after deficits in inhibition are seemingly corrected. The reviewers agreed that the main conclusion of enhanced synaptic excitation independent of inhibition is largely convincing and new. Enthusiasm for the topic and approach is tempered, however, by shared concerns about some weaknesses in data presentation, some claims that were not fully supported, missing controls and a need for a deeper analysis of the synaptic basis of enhanced excitation. These essential revisions will need to be addressed in a revised manuscript as outlined below.

Essential revisions:

1) Outcome measures from the ca^2+^ imaging experiments should be presented consistently across figures, and in a manner that clearly summarizes the differences between conditions (including blocking inhibition).

2) What is the circuit reorganization that generates larger EPSCs? Some additional insight using standard synaptic assays would provide mechanistic depth, and parallel experiments at the younger age would strengthen the conclusion about the developmental time course (which currently has conflicting data depending on the analysis shown in Figure 2).

3) Address whether fidelity of spike detection is affected by intact inhibition.

4) Either temper claims or provide additional data about intrinsic excitability of GCs and PVs.

5) Some experiments are missing controls, including a lack of counter-balance design in testing the ictogenicity of ChR2 stimulation, control groups in Figures4, 6, 7, and lack of characterization of the specificity and variability viral and Cre-based cell targeting. These limitations could be addressed by additional data and/or acknowledging limits of the interpretations. Likewise, conclusions in the abstract should be fully supported by the current data.

*Reviewer #1 (Recommendations for the authors):*

1) Figure 1 provides a nice validation for spike detection by ca^2+^ imaging using simultaneous cell-attached recordings. However, these experiments were performed in the presence of a GABAA antagonist rather than conditions mimicking the other imaging experiments (Figure 2). The authors should highlight in the text (rather than just methods) that these experiments are performed in an antagonist and also address whether block of GABAA receptors alters the fidelity of spike detection. This is important because subthreshold ca^2+^ transients can mimic spikes (i.e. Stocca et al., J Physiol 2007) and GABAA-mediated responses even in mature dentate GCs are typically depolarizing (Chiang et al., 2012).

2) In Figure 2, These are important and interesting results. However, how should the result in Figure 2B, columns 1-2 be interpreted? Does this mean there is a difference in the young mice (in contrast to the conclusion in the discussion)? If the difference relies on analyzing only the "best fields" it doesn't seem very reproducible. Rather, it seems that the PTX approach taken in Figure B3-B5 is far more informative, because the continued difference in in activation rules out the involvement of GABAA mediated inhibition (already ruling out feed-forward or feed-back inhibition suggested by subsequent experiments). This could be repeated in the early postnatal mice in place of the "best field analysis". But it is surprising that PTX appears to have very little effect on GC activation, for example, the blue lines in A3-5 and B3-5 appear nearly identical. It would be useful to have an additional summary analysis of "estimated activation" in each condition to highlight the magnitude of the differences (akin to panel 5D but related to %activation) to provide a clearer interpretation of the timing and magnitude of these effects.

My impression is also that the analysis in panel D5 underestimates magnitude of the differences between genotypes, which primarily generates greater recruitment of the active cell population rather than an increase in the number of spikes per active cell (this also seems more consistent with the interpretation of a lack of changes in intrinsic GC excitability). It is also not clear whether D5 represents both 1 and 4 pulses, postnatal and adult slices and PTX.

3). Based on data in Table 1, the authors conclude that altered intrinsic excitability does not contribute to spiking. However, this is not fully convincing – additional measures of intrinsic excitability that could contribute to afferent-induced spiking should be assessed, including AP threshold and measures related to integration of EPSPs including active dendritic potentials like Ih and membrane time constant. It would be useful if the authors measure APs during repetitive stimulation in GCs as in the interneuron recordings (Figure 3). Also, individual cell values are not typically averaged across mice – this should be corrected or justified.

4). The experiment with acute application of Hm1a is important, but the analysis of ca^2+^ does not seem comprehensive. The authors should provide analysis of the Estimated Activation (as in Figure 2A) to more strongly support the conclusion that the identified deficits in PV-IN spike generation and impairment in repetitive firing does not underlie the observed large-scale circuit impairment.

5) Along with Figure 2B3-5, Figure 5 is strong data supporting an increase in EPSCs in Scn1a slices, but the interpretation of the IPSCs is unclear. Is synaptic inhibition altered or not? The authors might want to avoid interpretation of the IPSCs without additional experiments to address it more convincingly.

More importantly the authors should address what underlies the larger EPSCs – is it more synapses, larger synapses, greater release probability? A more in-depth analysis would provide greater insight into this very interesting and unexpected finding.

6). There is huge variability in Figure 6C – possibly due to variability in ChR2 expression. How was this taken into account across mice? This is a similar concern for the interpretation of Figure 7 – how many SST interneurons were labeled versus PV interneurons? Did PV or SST interneuron activation have a different effect in WT mice? What is the specificity of the Cre lines? In general, it is well-established that activation of interneurons in DG suppresses spiking and seizure activity, so the results in Figure 7 don't add new understanding about the main novel finding of enhance excitatory transmission. This seems like a departure from the main topic so I'm not sure that it is necessary to perform all the relevant controls but rather the interpretations of these experiments might be tempered.

7) Some conclusions in the first paragraph of the discussion are oversimplifications that are not entirely consistent with the results. I.e. this abnormality was not present in early postnatal mice (P14-21) yet appeared during the chronic phase of the disorder. What about evoked EPSCs at the early age?

8) Some conclusions go beyond the present data. The idea that the identified circuit abnormality mirrors that seen in models of chronic temporal lobe epilepsy and suggests convergent mechanisms linking genetic and acquired causes of temporal is interesting and worthy of discussion. However, it seems beyond the scope of the current dataset to be included as a conclusion in the abstract. While the authors have identified hyperexcitation in the EC-DG circuit that results from greater synaptic excitation, they have not addressed whether similar dysfunction is present in other circuits nor whether this dysfunction in the EC-DG circuit is necessary for seizure activity in DS. This point might be worthy of discussion.

The statement in the abstract that excitability of parvalbumin interneurons (PV- INs) was normal and selective activation of PV-INs rescued circuit impairments is also an overstatement. Acute inhibition from either PVs or SSTs does not rescue circuit reorganization – it likely masks the effect of stronger excitatory drive. It is not clear that there is in fact, a circuit reorganization – the authors have just shown that EPSCs are larger which could mean more synapses, more receptors per synapse, or higher release probability – none of which would necessarily qualify as a "circuit reorganization".

*Reviewer #2 (Recommendations for the authors):*

This work would be much more impactful with experiments examining how and why the changes to synaptic signaling come about and how that translates into increased activation of DG. The major findings are of great interest, but the mismatch between the stimulation that reveals the differences in Figure 2 (1 or 4 synaptic inputs) and those used to test intrinsic properties Figure 3 and 4 limit the relevance of those intrinsic property experiments. While wild type control experiments often don't show differences (or in the case of Figure 7 may not show a response to change), leaving these experiments out suggests a lack of rigor in the design of the experiments and presentation of the data.

*Reviewer #3 (Recommendations for the authors):*

I would just like to expand on the criticism about the pharmacology.

1. Would it be possible to add to these results (Figure 4 and Figure 5) plots like those in figure 2?

2. More importantly – could you add the baseline dF/F0 to Figure 5F? I think it is necessary to show that picrotoxin causes an increase in dF/F0 in the control condition- just as a sanity check.

[Editors’ note: further revisions were suggested prior to acceptance, as described below.]

Thank you for resubmitting your work entitled "Corticohippocampal circuit dysfunction in a mouse model of Dravet syndrome" for further consideration by *eLife*. Your revised article has been evaluated by Gary Westbrook (Senior Editor) and a Reviewing Editor. The manuscript has been improved but there are some remaining issues that need to be addressed, as outlined below:

Essential revisions:

The reviewers agree this manuscript covers an important topic, provides novel data that will be of broad interest, and has gone to great lengths to address the comments of past reviewers. Please enhance the clarity of interpretations and presentation by addressing the following:

1) Tone down the strength of the conclusions related to pre- vs. postsynaptic locus based on the limitations specified in the comments below.

2) Include additional methodological details and clarifications as requested.

3) Strengthen the conclusions by moving essential data from the supplemental figures to main figures.

*Reviewer #1 (Recommendations for the authors):*

The authors have done a commendable job addressing the concerns raised by the reviewers. I have a few remaining points that should be addressed, but these do not reduce my enthusiasm for the significance of these results.

1. I am not very confident about conclusion that the dramatic increase in the EPSC is due to an increase in release probability for two reasons. First, the authors exclude the influence of post-synaptic component from the increased evoked EPSC using glutamate puffing experiments. But this is not the best approach to address this question due to the high variability/glutamate receptor desensitization, as well as ambiguity about whether synaptic vs extrasynaptic receptors mediate the response. A more conventional and sensitive approach is to compare mEPSC amplitude and frequency, or strontium-evoked uEPSCs, between Scn1a+/- and WT mice.

Second, the authors used the reduction in paired-pulse ratio (PPR) as a measure of higher release probability from Scn1a+/- PP synapses. While this is a reasonable interpretation, synapses from lateral and medial entorhinal cortex exhibit different PPR ratios that need to be taken into account. The uniformly high facilitation (PPR>1) shown in WT slices in Figure 5 H and I is indicative of transmission from the lateral perforant path, whereas the variable (PPR=1) in the other panels including the wt slices in Figure 5 Supp 1D is indicative of transmission from a mixed population of lateral and medial synapses (see Petersen et al., Neuroscience 2013). Interpretation of PPR changes in DG requires identification of the pathway stimulated. But since the pre vs postsynaptic locus is not essential to the main conclusions, this concern could be address by toning down the conclusion and discussing these caveats. Also, the glutamate-evoked currents should not be called EPSCs in the text and legends.

2. Further clarification would be beneficial for Figure 5 —figure supplement 2 panel A, where authors showed a statistically significant difference in magnitude of evoked GC responses between Scn1a+/- and WT using ca^2+^ imaging. However, panel C-F and title indicate no difference.

3. Finally, figure 3A and 4A require better representative traces with reduced line width. It is helpful that the authors use the same color scheme consistently, and it would be even easier to have the colors labeled within the figures instead of in the legend.

---

## [Author Response]

Essential revisions:1) Outcome measures from the ca^2+^ imaging experiments should be presented consistently across figures, and in a manner that clearly summarizes the differences between conditions (including blocking inhibition).

Thank you for this helpful feedback. We have collected an entirely new dataset at the early postnatal timepoint and now include data on the effect of blocking inhibition (via picrotoxin) consistently at both timepoints (Figure 2; Figure 2 —figure supplement 1). In this revised manuscript we have improved the consistency of presentation of the calcium imaging data. Please note however that we conceptualize this imaging data as fitting into two categories, which do require a slightly different graphical depiction:

– Unpaired data in which we analyze responses across a range of stimulation conditions (Figure 2 and associated supplemental Figure 2 —figure supplement 1 and Figure 2 —figure supplement 3)

– Paired data in which we assess the response within a given imaging field to a manipulation performed at a single stimulation condition (Hm1a data in Figure 4 and Figure 4 —figure supplement 1; PTX data in Figure 5 and Figure 5 —figure supplement 2; PV-IN data that appears in Figure 8)

2) What is the circuit reorganization that generates larger EPSCs? Some additional insight using standard synaptic assays would provide mechanistic depth, and parallel experiments at the younger age would strengthen the conclusion about the developmental time course (which currently has conflicting data depending on the analysis shown in Figure 2).

We agree that this was under-explored in the initial submission. Our revised manuscript includes substantial new data to address this critique, including:

– A parallel characterization in juvenile mice of the intrinsic properties of granule cells and PV cells in wild-type and *Scn1a*+/- mice

– Isolation of the post-synaptic component of the response in granule cells via local application of glutamate

– Determination of the EPSC paired-pulse ratio to measure pre-synaptic release probability to provide evidence in support of a pre- vs. post-synaptic mechanism.

Together, these findings are aligned to suggest that the larger EPSCs are driven by a presynaptic mechanism. This finding implicates dysregulation of the excitatory input from entorhinal cortex to dentate gyrus, further investigation of which is beyond the scope of this manuscript and what could reasonably be achieved in the time allotted for resubmission.

3) Address whether fidelity of spike detection is affected by intact inhibition.

We agree with Reviewer #1 and this is an important point. We now include additional data collected in the absence of picrotoxin to address this concern (Figure 1H) and have expanded relevant discussion in the text. We confirm that detection of single GC spikes is retained with inhibition intact.

4) Either temper claims or provide additional data about intrinsic excitability of GCs and PVs.

We now provide additional data on and a much more comprehensive analysis of intrinsic properties of GCs and PV-INs at both developmental timepoints (Tables 1 and 2).

5) Some experiments are missing controls, including a lack of counter-balance design in testing the ictogenicity of ChR2 stimulation, control groups in Figures4, 6, 7, and lack of characterization of the specificity and variability viral and Cre-based cell targeting. These limitations could be addressed by additional data and/or acknowledging limits of the interpretations. Likewise, conclusions in the abstract should be fully supported by the current data.

Thank you for this very helpful feedback, which has improved the transparency and rigor of our manuscript.

– For the Hm1a experiment (Figure 4), we now present wild-type control data for both PV electrophysiology and 2P circuit-level imaging (Figure 4 —figure supplement 1).

– Despite what we perceive as the novelty of this optogenetic stimulation approach, we have elected to remove the optogenetic imaging data (previously Figure 6C), as we indeed found results to be highly variable across mice and between slices from an individual mouse.

– As previously designed, the entorhinal cortex ictogenicity experiment could not be performed with counter-balancing due to a kindling-like effect. We therefore fully redesigned the experiment as a single-day experiment with relevant controls for both photostimulation and genotype (now Figure 6).

– We have added an in vivo experiment in which we chemogenetically inhibit dentate PV cells and demonstrate a lower temperature threshold for seizure generation. This experiment was performed with a counter-balanced design and with controls for genotype and for CNO.

– For the experiment demonstrating a decrease in circuit activation in response to PV stimulation (now Figure 8), we were not able to perform a wild-type control due to very low levels of granule cell activation in wild-type mice under those conditions (see Figure 2, panel A3 – response to 1 pulse in young adult wild-type mice). In other words, in the wild-type mice, there was essentially no signal to block. This is consistent with previously published data showing low recruitment of dentate gyrus granule cells in response to perforant path activation (e.g., Yu et al., 2013 J Neurosci; Ewell and Jones 2010 J Neurosci; Dieni et al., 2013 J Neurosci; etc.). However, in this experiment, we in fact conceptualize the SST activation as the control group (for the PV activation), which we have clarified in the text.

– We now include data to characterize the PV- and SST-Cre based cell targeting (Figure 8; Figure 8 —figure supplement 1).

Reviewer #1 (Recommendations for the authors):1) Figure 1 provides a nice validation for spike detection by ca^2+^ imaging using simultaneous cell-attached recordings. However, these experiments were performed in the presence of a GABAA antagonist rather than conditions mimicking the other imaging experiments (Figure 2). The authors should highlight in the text (rather than just methods) that these experiments are performed in an antagonist and also address whether block of GABAA receptors alters the fidelity of spike detection. This is important because subthreshold ca^2+^ transients can mimic spikes (i.e. Stocca et al., J Physiol 2007) and GABAA-mediated responses even in mature dentate GCs are typically depolarizing (Chiang et al., 2012).

We appreciate this important point. We have added clarity to the main text as suggested regarding the use of picrotoxin in the validation experiment and we have also added the above references to the text. To address the possibility of subthreshold calcium transients resulting in “false positive” spike detection, we have performed a separate experiment in which we measured calcium signals in the absence of picrotoxin, focusing on cells that fired either 0 or 1 action potentials (Figure 1H). We confirmed that we were still able to resolve single action potentials.

2) In Figure 2, These are important and interesting results. However, how should the result in Figure 2B, columns 1-2 be interpreted? Does this mean there is a difference in the young mice (in contrast to the conclusion in the discussion)? If the difference relies on analyzing only the "best fields" it doesn't seem very reproducible. Rather, it seems that the PTX approach taken in Figure B3-B5 is far more informative, because the continued difference in in activation rules out the involvement of GABAA mediated inhibition (already ruling out feed-forward or feed-back inhibition suggested by subsequent experiments). This could be repeated in the early postnatal mice in place of the "best field analysis". But it is surprising that PTX appears to have very little effect on GC activation, for example, the blue lines in A3-5 and B3-5 appear nearly identical. It would be useful to have an additional summary analysis of "estimated activation" in each condition to highlight the magnitude of the differences (akin to panel 5D but related to %activation) to provide a clearer interpretation of the timing and magnitude of these effects.

We thank Reviewer #1 for these helpful comments.

– First, we agree that the picrotoxin approach was preferable to the perhaps somewhat arbitrary “best fields analysis”, and we have therefore discarded this “best fields analysis” and collected a new dataset at the early postnatal timepoint, including a picrotoxin wash-on condition for each imaging field. Figure 2 is now updated with these results, and the conclusion is essentially unchanged. (Note that Figure 1A1-2, C1-2, and D1-2 are pooled across the two datasets which were very similar, whereas B1-2 includes only this new dataset.). This new presentation of data achieves the standardization requested by the Reviewers mentioned above.

– Second, to explain why row B (only those cells that *do* respond to maximal stimulation in the presence of PTX) appears qualitatively similar to row A (all identified cells), we now include a supplemental figure quantifying the percentage of cells activated by picrotoxin in each imaging field (Figure 2 —figure supplement 2). We note that this is a very high percentage (~88-96%); in other words, the large majority of the cells do fire in response to a strong stimulus in the presence of picrotoxin. We have clarified this in the text as well.

– Third, we agree that the continued genotype difference in activation in the presence of PTX is an important finding. We now show expanded analysis (Figure 5E) to highlight this result.

My impression is also that the analysis in panel D5 underestimates magnitude of the differences between genotypes, which primarily generates greater recruitment of the active cell population rather than an increase in the number of spikes per active cell (this also seems more consistent with the interpretation of a lack of changes in intrinsic GC excitability). It is also not clear whether D5 represents both 1 and 4 pulses, postnatal and adult slices and PTX.

This is an interesting point. Indeed, Figure 2D5-6 answers the question “How activated were the activated cells?” (i.e., how many spikes were there per active cell). We thus only included cells that were activated under each individual stimulation condition (such that the “*n*” varies across the x-axis). However, per Figure 2A-C we know that there was also a genotype difference in the proportion of activated cells. We have thus added a supplemental figure (Figure 2 —figure supplement 3) designed to answer the different question: “How much activation was there overall?” (i.e., how many spikes were there across the population). In this new supplemental figure, we include all cells that respond to the maximal stimulation condition (no PTX); i.e., averaging many 0 values in the lower stimulation conditions. We have updated the text to make this distinction clear. (Overall, the genotype findings persist across these analysis conditions.)

3). Based on data in Table 1, the authors conclude that altered intrinsic excitability does not contribute to spiking. However, this is not fully convincing – additional measures of intrinsic excitability that could contribute to afferent-induced spiking should be assessed, including AP threshold and measures related to integration of EPSPs including active dendritic potentials like Ih and membrane time constant. It would be useful if the authors measure APs during repetitive stimulation in GCs as in the interneuron recordings (Figure 3). Also, individual cell values are not typically averaged across mice – this should be corrected or justified.

We have updated both tables to include a more comprehensive set of parameters, and no longer average across mice. We now present data from repetitive stimulation of GCs at both timepoints, which shows no significant genotype differences (Figure 2 —figure supplement 4).

4). The experiment with acute application of Hm1a is important, but the analysis of ca^2+^ does not seem comprehensive. The authors should provide analysis of the Estimated Activation (as in Figure 2A) to more strongly support the conclusion that the identified deficits in PV-IN spike generation and impairment in repetitive firing does not underlie the observed large-scale circuit impairment.

We have repeated this experiment and now present both proportional activation and amplitude data, for both *Scn1a*+/- (Figure 4), and wild-type (now Figure 4).

5) Along with Figure 2B3-5, Figure 5 is strong data supporting an increase in EPSCs in Scn1a slices, but the interpretation of the IPSCs is unclear. Is synaptic inhibition altered or not? The authors might want to avoid interpretation of the IPSCs without additional experiments to address it more convincingly.

We now include EPSC/IPSC data recorded from the early postnatal timepoint as well. We agree that the IPSC data is less straightforward: at both timepoints we do find a very slight increase in the magnitude of the IPSC in *Scn1a*+/- mice, reaching statistical significance as determined by comparison of curve fits, but without significant differences at the individual data points. Overall, we agree that it is not clear whether this difference is physiologically meaningful. We do now mention this difference in the Discussion in reference to the early postnatal dataset, highlighting that, although PV-INs fail much sooner than wild-type, their higher input resistance and slightly left-shifted *I-*F curve (now Figure 3) complicates interpretation of their function within the circuit.

More importantly the authors should address what underlies the larger EPSCs – is it more synapses, larger synapses, greater release probability? A more in-depth analysis would provide greater insight into this very interesting and unexpected finding.

We address this important point as follows:

– We performed two experiments in the young adult timepoint: (1) We measured the response to puffed glutamate (to eliminate the presynaptic component) and (2) we determined the paired pulse ratio (PPR) from repetitive stimulation (to assess release probability). As shown in Figure 5 and Figure 5 —figure supplement 1, we found no difference in the response to puffed glutamate, but a significantly lower PPR in the *Scn1a*+/- mice to suggest that indeed the release probability is higher in *Scn1a*+/- mice.

– We further measured EPSCs and IPSCs in the juvenile timepoint (Figure 5 —figure supplement 2) and found no increase in EPSCs in *Scn1a*+/- mice at the juvenile time point, as well as no genotype difference in release probability.

6). There is huge variability in Figure 6C – possibly due to variability in ChR2 expression. How was this taken into account across mice? This is a similar concern for the interpretation of Figure 7 – how many SST interneurons were labeled versus PV interneurons? Did PV or SST interneuron activation have a different effect in WT mice? What is the specificity of the Cre lines? In general, it is well-established that activation of interneurons in DG suppresses spiking and seizure activity, so the results in Figure 7 don't add new understanding about the main novel finding of enhance excitatory transmission. This seems like a departure from the main topic so I'm not sure that it is necessary to perform all the relevant controls but rather the interpretations of these experiments might be tempered.

We agree with these critiques, which we have addressed as follows:

– Regarding Figure 6C, as noted above, we performed additional experiments with optogenetic stimulation of entorhinal cortex and 2P imaging in DG slice, but found a continued high degree of variability within and across mice; we also observed variability even within an imaging field, with difficulty reproducing the same response across trials of identical stimulation. Hence, we consider that further work is required to further refine the approach and decided to remove this data from the manuscript, despite the perceived novelty and promise of this type of experiment.

– We attempted to test the response of PV and SST activation in WT mice but found that the baseline signal was too low to perform the experiment. (Note that this experiment was performed with a single pulse at 500 µA, which activates very few cells in the WT condition; see Figure 2 panel A3.) Overall, we conceptualize SST as a control for the PV experiment (within the *Scn1a*+/- mouse), which we have clarified in the text.

– We in fact observed *more* ChrimsonR-labeled SST cells than PV cells per field (quantified in the Results section), although (as we note in the Discussion section), our stimulation will activate both cell bodies and fibers, making a true quantitative comparison challenging.

– We now address Cre-line specificity with immunostaining (Figure 8 —figure supplement 1) as well as differential anatomic localization of ChrimsonR-expressing fibers (Figure 8).

7) Some conclusions in the first paragraph of the discussion are oversimplifications that are not entirely consistent with the results. I.e. this abnormality was not present in early postnatal mice (P14-21) yet appeared during the chronic phase of the disorder. What about evoked EPSCs at the early age?

We now present data at the early postnatal timepoint (Figure 5 —figure supplement 2), with no difference seen in E/I ratio, EPSC magnitude, or PPR. We have edited the Discussion section accordingly.

8) Some conclusions go beyond the present data. The idea that the identified circuit abnormality mirrors that seen in models of chronic temporal lobe epilepsy and suggests convergent mechanisms linking genetic and acquired causes of temporal is interesting and worthy of discussion. However, it seems beyond the scope of the current dataset to be included as a conclusion in the abstract. While the authors have identified hyperexcitation in the EC-DG circuit that results from greater synaptic excitation, they have not addressed whether similar dysfunction is present in other circuits nor whether this dysfunction in the EC-DG circuit is necessary for seizure activity in DS. This point might be worthy of discussion.

We have edited the Abstract to remove this verbiage, and have included consideration of this point in the Discussion section.

The statement in the abstract that excitability of parvalbumin interneurons (PV- INs) was normal and selective activation of PV-INs rescued circuit impairments is also an overstatement. Acute inhibition from either PVs or SSTs does not rescue circuit reorganization – it likely masks the effect of stronger excitatory drive. It is not clear that there is in fact, a circuit reorganization – the authors have just shown that EPSCs are larger which could mean more synapses, more receptors per synapse, or higher release probability – none of which would necessarily qualify as a "circuit reorganization".

Agreed. We have edited the abstract accordingly.

Reviewer #2 (Recommendations for the authors):This work would be much more impactful with experiments examining how and why the changes to synaptic signaling come about and how that translates into increased activation of DG. The major findings are of great interest, but the mismatch between the stimulation that reveals the differences in Figure 2 (1 or 4 synaptic inputs) and those used to test intrinsic properties Figure 3 and 4 limit the relevance of those intrinsic property experiments. While wild type control experiments often don't show differences (or in the case of Figure 7 may not show a response to change), leaving these experiments out suggests a lack of rigor in the design of the experiments and presentation of the data.

These recommendations have been addressed in detail above.

Reviewer #3 (Recommendations for the authors):I would just like to expand on the criticism about the pharmacology.1. Would it be possible to add to these results (Figure4 and Figure 5) plots like those in figure 2?

See the response above.

2. More importantly – could you add the baseline dF/F0 to Figure 5F? I think it is necessary to show that picrotoxin causes an increase in dF/F0 in the control condition- just as a sanity check.

We now include baseline d*F*/*F0* data for both developmental timepoints (Figure 5; Figure 5 —figure supplement 2).

[Editors' note: further revisions were suggested prior to acceptance, as described below.]

Essential revisions:The reviewers agree this manuscript covers an important topic, provides novel data that will be of broad interest, and has gone to great lengths to address the comments of past reviewers. Please enhance the clarity of interpretations and presentation by addressing the following:1) Tone down the strength of the conclusions related to pre- vs. postsynaptic locus based on the limitations specified in the comments below.

We have toned down the strength of our conclusions related to the pre- vs. postsynaptic locus (glutamate puff experiment and analysis of PPR) as required. We have added mention of the limitations of these experiments as noted by both Reviewers.

2) Include additional methodological details and clarifications as requested.

Done. Please see point by point details below.

3) Strengthen the conclusions by moving essential data from the supplemental figures to main figures.

Done. Please see new versions of Figure 3 and Figure 4, each of which now incorporates data previously displayed in an associated Figure —figure supplement.

Reviewer #1 (Recommendations for the authors):The authors have done a commendable job addressing the concerns raised by the reviewers. I have a few remaining points that should be addressed, but these do not reduce my enthusiasm for the significance of these results.1. I am not very confident about conclusion that the dramatic increase in the EPSC is due to an increase in release probability for two reasons. First, the authors exclude the influence of post-synaptic component from the increased evoked EPSC using glutamate puffing experiments. But this is not the best approach to address this question due to the high variability/glutamate receptor desensitization, as well as ambiguity about whether synaptic vs extrasynaptic receptors mediate the response. A more conventional and sensitive approach is to compare mEPSC amplitude and frequency, or strontium-evoked uEPSCs, between Scn1a+/- and WT mice.Second, the authors used the reduction in paired-pulse ratio (PPR) as a measure of higher release probability from Scn1a+/- PP synapses. While this is a reasonable interpretation, synapses from lateral and medial entorhinal cortex exhibit different PPR ratios that need to be taken into account. The uniformly high facilitation (PPR>1) shown in WT slices in Figure 5 H and I is indicative of transmission from the lateral perforant path, whereas the variable (PPR=1) in the other panels including the wt slices in Figure 5 Supp 1D is indicative of transmission from a mixed population of lateral and medial synapses (see Petersen et al., Neuroscience 2013). Interpretation of PPR changes in DG requires identification of the pathway stimulated. But since the pre vs postsynaptic locus is not essential to the main conclusions, this concern could be address by toning down the conclusion and discussing these caveats. Also, the glutamate-evoked currents should not be called EPSCs in the text and legends.

We appreciate these helpful comments:

– Conclusions based on the glutamate puff experiments have been toned down in both the Results and Discussion section, and the ambiguity of synaptic versus extrasynaptic receptor activation has been acknowledged.

– Interpretation of the meaning of the change in PPR has also been toned down, and the important caveat as to the proportionate activation of lateral versus medial perforant path has been added to the Discussion section, along with the Peterson et al., reference suggested by Reviewer #1.

– The term “EPSC” has been replaced in all references to exogenous glutamate puff-evoked currents.

2. Further clarification would be beneficial for Figure 5 —figure supplement 2 panel A, where authors showed a statistically significant difference in magnitude of evoked GC responses between Scn1a+/- and WT using ca^2+^ imaging. However, panel C-F and title indicate no difference.

We appreciate that Reviewer #1 has pointed this out. Indeed, as shown in Figure 2D1, there is a small but significant difference in d*F*/*F0* at this juvenile timepoint. We find that the genotype difference persists in the presence of PTX (Figure 5 —figure supplement 2A) and is therefore independent of synaptic inhibition. However, we did not find evidence of increased synaptic excitation onto GCs at this timepoint (Figure 5 —figure supplement 2) and, thus, the mechanism of this finding remains to be fully clarified. We have now highlighted this in the text.

3. Finally, figure 3A and 4A require better representative traces with reduced line width. It is helpful that the authors use the same color scheme consistently, and it would be even easier to have the colors labeled within the figures instead of in the legend.

We appreciate these helpful comments towards improving the manuscript.

– Representative traces in Figure 3 have been modified (expanded on the horizontal axis) such that individual action potentials are now visible.

– Explicit color labeling has been added to those Figures for which it is not already provided using different color(s) of text, namely:

Figure 1; Figure 2 and associated Figure 2 —figure supplements; Figure 5 —figure supplement 1 and 2

– However, for Figure 4A, even with maximally reduced line width it would not be possible to resolve individual action potentials without greatly increasing the size of the traces. Since the purpose of this figure panel is to qualitatively illustrate the spike rundown and eventual failures at baseline and correction after addition of Hm1a, we believe that the current formatting is appropriate. We have added a comment in the text along these lines.